# Neural mechanisms of flexible perceptual inference

John Schwarcz[1], Jan Bauer[1,2], Haneen Rajabi[1], Gabrielle Marmur[1], Robert Reiner[1], Eran Lottem[1‡], Jonathan Kadmon[1‡*]

**1** Edmond and Lily Safra Center for Brain Sciences (ELSC), The Hebrew University of Jerusalem, Jerusalem, Israel, **2** Gatsby Computational Neuroscience Unit, University College London, London, United Kingdom

☉ These authors contributed equally to this work.
‡ EL and JK also contributed equally to this work.
* jonathan.kadmon@mail.huji.ac.il

## Abstract

What seems obvious in one context can take on an entirely different meaning if that context shifts. While context-dependent inference has been widely studied, a fundamental question remains: how does the brain simultaneously infer both the meaning of sensory input and the underlying context itself, especially when the context is changing? Here, we study flexible perceptual inference—the ability to adapt rapidly to implicit contextual shifts without trial and error. We introduce a novel change-detection task in dynamic environments that requires tracking latent state and context. We find that mice exhibit first-trial behavioral adaptation to latent context shifts driven by inference rather than reward feedback. By deriving the Bayes-optimal policy under a partially observable Markov decision process, we show that rapid adaptation emerges from sequential updates of an internal belief state. In addition, we show that artificial neural networks trained via reinforcement learning achieve near-optimal performance, implementing Bayesian inference-like mechanisms within their recurrent dynamics. These networks develop flexible internal representations that enable adaptive inference in real-time. Our findings establish flexible perceptual inference as a core principle of cognitive flexibility, offering computational and neural-mechanistic insights into adaptive behavior in uncertain environments.

## Author Summary

How do animals adapt their behavior when the rules of their environment change unexpectedly? In this study, we address this fundamental question by training mice to perform a task in which they must infer not only the current state of their environment, but also the context that determines how reliable their sensory cues are. We show that mice quickly adjust their decisions to changes in both state

**Data availability statement:** All data including behavioral data, simulation code for neural-network modelling, and analysis scripts, is publicly available at https://github.com/kadmon-lab/flexible-perceptual-inference.

**Funding:** This work was supported by The Gatsby Charitable Foundation (J.K.; https://www.gatsby.org.uk), the National Institute of Psychobiology in Israel (NIPI; J.K.; https://psychobiology.org.il), and the Israel Science Foundation (ISF; E.L.; grant no. 1269/20; https://www.isf.org.il). The funders had no role in the study design, data collection and analysis, the decision to publish, or the preparation of the manuscript.

**Competing interests:** The authors have declared that no competing interests exist.

and context, often before receiving any explicit feedback. To understand how such flexible inference could be implemented in the brain, we developed a computational framework based on Bayesian inference and trained artificial neural networks to solve the same task. Our analysis reveals that both mice and neural networks learn to infer hidden contexts from sensory inputs, relying on internal computations rather than just trial-and-error learning. We also demonstrate that the internal dynamics of the networks mirror the complex, context-dependent strategies predicted by optimal Bayesian theory. These findings shed light on the computational principles and potential neural mechanisms that allow animals to flexibly adapt their behavior in uncertain and changing environments.

# 1 Introduction

As dusk settles over a field, a mouse hesitates at the edge of its burrow, torn between hunger and caution. Every rustle, shadow, and scent carries ambiguity—harmless background noise or a lurking predator? A flicker in the sky—owl or drifting cloud? Each cue is uncertain, yet survival hinges on interpreting them swiftly and accurately. Crucially, the mouse's decisions are shaped not only by immediate evidence, but also by an inferred context: is the wind in the underbrush stronger than usual, making rustling sounds less reliable? Are the shadows deeper tonight, obscuring movement? Rather than passively accumulating sensory evidence, the mouse must continuously update its internal model of the world, refining both its beliefs about what is happening and how incoming information should be weighted and interpreted.

This dual challenge—extracting meaning from uncertain inputs while simultaneously inferring the latent structure that shapes their interpretation—is fundamental to perception and decision-making. The brain must solve it efficiently and continuously. This process, which we term **flexible perceptual inference**, enables organisms to rapidly adapt to changing conditions, resolve uncertainty, and make informed decisions without relying solely on trial-and-error learning. This cognitive flexibility is as crucial for a mouse navigating the wilderness as it is for any intelligent system operating in a complex and unpredictable world [1].

The challenge of probabilistic reasoning in uncertain environments is best captured by Bayesian decision theory, which provides a principled framework for perceptual inference. This framework formalizes how optimal agents update their beliefs in response to uncertain observations [2–4]. Empirical studies have shown that human and animal behavior is often aligned with Bayesian inference, particularly in perceptual decision-making tasks where organisms integrate sequential sensory evidence to make decisions under uncertainty [5–8].

A common framework for modeling the accumulation of perceptual evidence is the drift-diffusion model (DDM) [9–12], which approximates Bayesian inference under different conditions [13,14]. However, traditional DDM formulations assume a fixed and known context, leaving open the question of how real-time inference adapts to latent

contextual uncertainty. Much of the existing work on learning in changing environments has focused on how organisms detect and adjust to changes in volatility [15–17], often by modifying learning rates or belief update rules in hierarchical Bayesian models [18–20].

Existing studies on contextual evidence integration typically fall into three main categories. The first examines how neural circuits and artificial networks perform context-dependent integration but assume an explicit externally provided context signal [21–24]. The second asks how humans and animals adapt decision making in dynamic environment by discounting old evidence [25–28]. The third category considers contextual inference, where the context must be inferred from observations [29–31]. However, the mechanisms that allow the brain to infer an implicit context and adapt the dynamics of evidence accumulation simultaneously remain poorly understood [32].

Crucially, distinguishing between these mechanisms is challenging, as behavior alone does not uniquely constrain the underlying neural computations [33–37]. Multiple evidence accumulation models can generate behavior that appears consistent with Bayesian inference, making it difficult to infer how the brain implements flexible perceptual inference. Resolving this ambiguity requires moving beyond behavioral signatures to examine how neural circuits represent, update, and adapt decision variables in response to changing contexts [38–41].

To address these open questions, we investigate how neural circuits perform *flexible perceptual inference*—how they simultaneously infer both latent states and context from ambiguous sensory input and dynamically adjust evidence integration. We introduce a change-detection task in which both state and context must be inferred from a single stream of inputs in a dynamic and uncertain environment. Using this framework, we make three key contributions: (1) We show that mice rapidly adjust their behavioral strategy in response to implicit changes in input statistics, adapting within a single trial without reward or feedback. This suggests that their behavior is guided by a learned inference model rather than trial-and-error learning. (2) We derive the Bayes-optimal policy for the task, demonstrating that optimal inference requires a nontrivial real-time adaptation of evidence integration, in which both accumulation rates and decision boundaries adjust dynamically. (3) We show that artificial recurrent neural networks trained via reinforcement learning achieve near-optimal performance by embedding the sequential update of the Bayes-optimal decision variable within their recurrent dynamics. Together, our findings establish flexible perceptual inference as a fundamental computational principle, providing mechanistic insights into how biological and artificial systems adapt to uncertain and dynamically changing environments.

## 2 A change-detection task to study flexible perceptual inference

To investigate how agents infer both an evolving state and a shifting context from the same inputs, we designed a **change-detection task with variable sensory reliability**. The task requires tracking a latent state that determines reward availability while simultaneously adapting to changes in sensory reliability, which we consider the "context". This simple dual-inference problem reflects the challenges of more general real-world problems.

**Task structure.** The agent must infer a latent binary state, $s_t \in \{0, 1\}$, where $s_t = 0$ denotes an *unsafe* state and $s_t = 1$ a *safe* state. The state evolves probabilistically: each trial begins in the unsafe state, transitioning to safe with probability $\lambda$ per time step. The agent's objective is to act ($a_t = 1$) only in a safe state to obtain a reward and to withhold action ($a_t = 0$) otherwise. A trial ends immediately upon action.

The agent does not directly observe $s_t$ but instead receives a binary input, $x_t \in \{0, 1\}$, which we denote as *nogo* ($x_t = 0$) or *go* ($x_t = 1$). These observations are **noisy reflections of the current state**. A latent context parameter, $\theta_t$, determines sensory reliability: in an unsafe state, the input is veridical ($x_t = 0$) with probability $1 - \theta_t$ and misleading ($x_t = 1$) with probability $\theta_t$. In a safe state, the observations are always reliable ($x_t = 1$). The context parameter $\theta_t$ changes randomly between trials, grouping them into blocks of varying length. The trial and block structure of the task is depicted in Fig 1.

**Optimal strategy and speed-accuracy trade-off.** The agent faces a fundamental trade-off between acting prematurely (risking incorrect responses) and delaying too long (reducing the reward rate [10,42,43]). The optimal policy is characterized by a **waiting time**, defined as the number of consecutive *go* signals since the last *nogo*. This waiting time

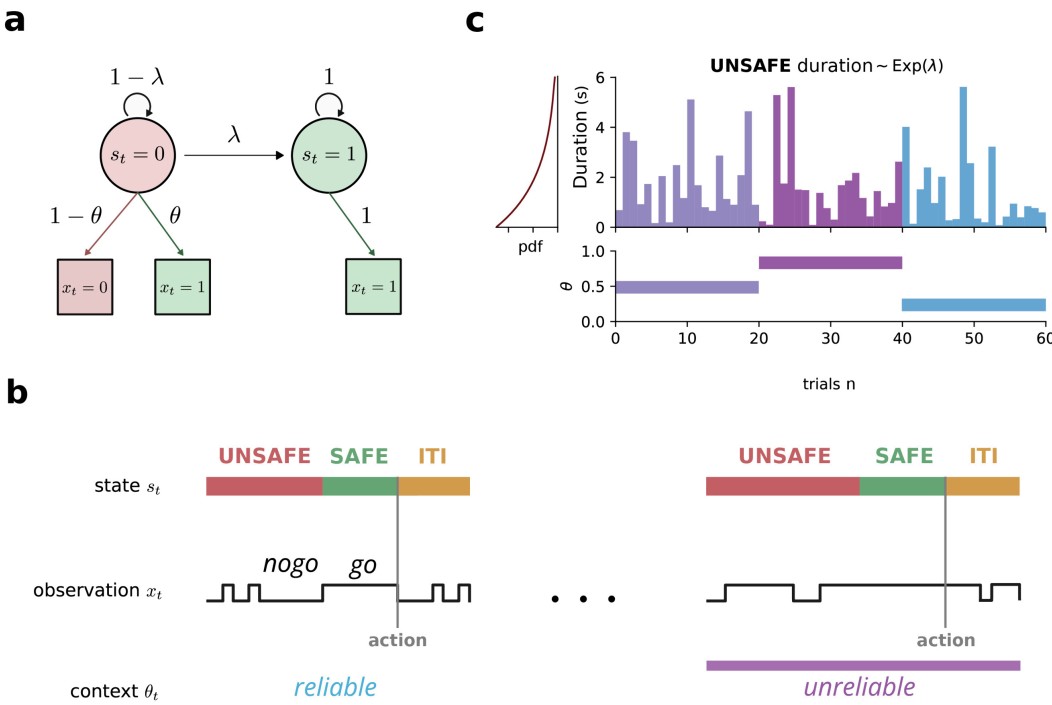

**Fig 1**. **Change-detection task requiring flexible perceptual inference. (a)** Task structure. Each trial begins in an *unsafe* state ($s_t = 0$) that transitions to a *safe* state ($s_t = 1$) with probability $\lambda$ per time step. The agent observes binary signals: *nogo* ($x_t = 0$) or *go* ($x_t = 1$). In the safe state, observations are reliable (*go* with probability 1). In the unsafe state, observations depend on context parameter $\theta$: agents receive misleading *go* signals with probability $\theta$ and veridical *nogo* signals with probability $1 - \theta$. Actions terminate trials immediately. **(b)** Reward structure and speed-accuracy trade-off. Agents earn rewards only for acting in the safe state. Premature actions in the unsafe state yield no reward and incur time penalties. The optimal strategy requires waiting for sufficient consecutive *go* signals to ensure state transition, with waiting time depending on the reliability context $\theta$. Inter-trial intervals (ITIs) have uniform duration and *unsafe*-like statistics, preventing anticipation of trial onset. **(c)** Block design with latent context changes. The context parameter $\theta$ remains constant within blocks but changes unpredictably between blocks, creating periods of *reliable* (low $\theta$) and *unreliable* (high $\theta$) sensory evidence. Following block switches, optimal performance requires jointly inferring both the current state and the new context from the observation stream alone, without external cues or feedback.

should depend on $\theta_t$: as sensory reliability decreases (higher $\theta_t$), the agent should wait longer to ensure that the state transition has occurred. Maximizing the reward rate over time thus imposes a **speed-accuracy trade-off**, balancing cautious waiting with efficient action. For each context $\theta$, there exists a finite optimal waiting time $\tau^\star(\theta)$ that maximizes the reward rate $r(\tau; \theta)$.

**The challenge of ambiguous inputs.** A key difficulty in this task is resolving signal ambiguity. Let us consider a long string of consecutive *go* signals. This abnormal observation could indicate either that the state has transitioned to safe or that the context has shifted to a less reliable environment ($\theta$ increasing). To act optimally, the agent must **jointly infer the latent state and the context**—a form of flexible perceptual inference. Unlike traditional reinforcement learning, where adaptation is driven by trial-and-error, optimal behavior requires real-time inference within a single trial.

**Implications for flexible inference.** To perform well, the agent must develop internal belief states that integrate incoming observations with prior expectations about state transitions and context shifts. In Sect 3.1, we show that **mice trained on this task adapt their behavior within the first trial of a new context**, indicating that they infer context implicitly from observations rather than relying on reward feedback, motivating our study of flexible perceptual inference. In Sect 3.2, we derive the **Bayesian-optimal solution** and analyze the dynamics of an ideal observer. Finally, in Sects 3.4 and 3.5, we demonstrate that artificial neural networks trained on this task develop representations that mirror Bayesian inference.

## 3 Results

### 3.1 Trained mice adapt their behavior to a new context within the first trial

We trained adult mice in a physical implementation of our task. Water-deprived, head-fixed mice were placed in front of a water delivery spout (Fig 2A). In the experiments, we represented the *go* signals ($x_t = 1$) as a continuous auditory tone, while the *nogo* signals ($x_t = 0$) were represented by silence. The sound sequences were generated using the trial structure described in Sect 2. Each signal lasted 0.2 seconds, with no gaps between consecutive signals. The mice were rewarded with a water drop for successful actions, defined as licking the water delivery port during the safe state. Detailed experimental procedures are provided in the Methods section.

To assess how mice adapt their behavior to different latent states of the environment, we trained and tested a group of 17 mice on our task, switching between blocks of varying values of $\theta$. We focused on analyzing the mice's *waiting times*, defined as the duration between the last *nogo* signal and the first lick. Since the underlying state is hidden, the number of consecutive *go* cues is therefore the only relevant observable. In the next section, we derive the optimal solution in terms of waiting times.

Fig 2B presents the average waiting times for different contexts, demonstrating that, consistent with normative predictions we detail in Sect 3.2, mice wait longer as $\theta$ increases (Pearson's r = 0.83, $p < .005$). Although behavioral statistics show that mice modulate their waiting time given the latent state, they do not fully reveal what elicited this behavioral

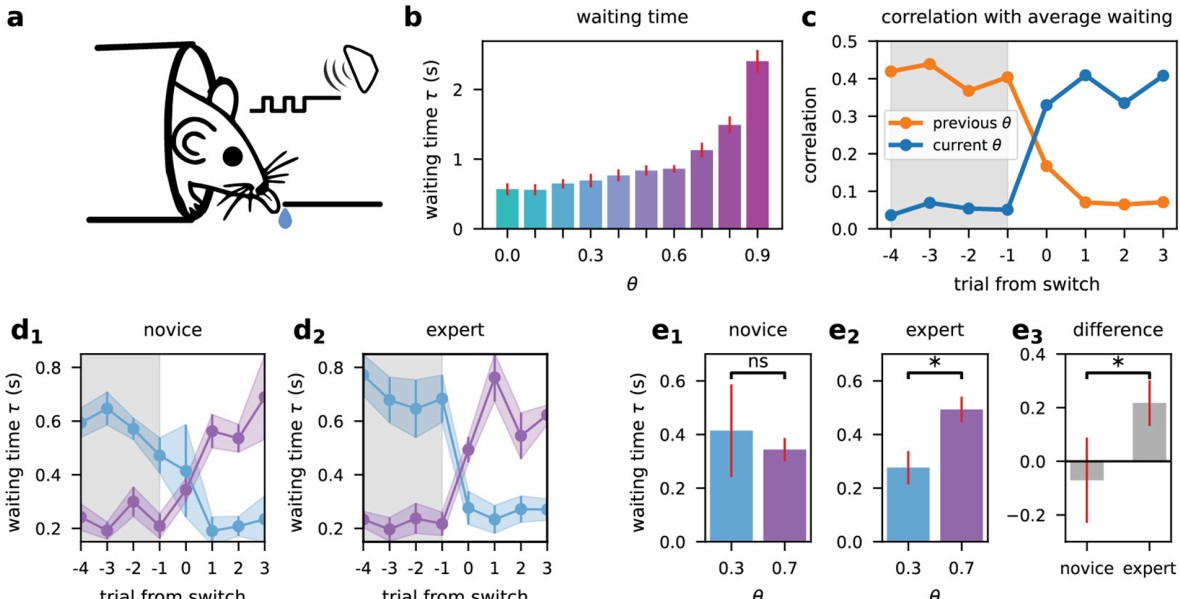

**Fig 2. Mice adapt waiting times to uncertainty without explicit feedback. (a)** Experimental setup. Head-fixed, water-restricted mice performed an auditory version of the task. Licking during the *safe* state delivered water reward, requiring mice to infer latent states over time. **(b)** Mean waiting time $\tau$ as a function of the contextual parameter $\theta$. Waiting times were longer in higher-uncertainty contexts. **(c)** Correlation between trial-by-trial waiting times and average waiting times from the previous (orange) and current (blue) context blocks. Following a block switch (shaded region), mice rapidly adjusted waiting times to the new context, showing **first-trial adaptation** based on inference rather than feedback. **(d)** Trial-by-trial waiting times aligned to block transitions in **novice** (first five sessions per mouse, left) and **expert** (last five sessions per mouse, right) animals. Data are shown separately for transitions from high to low uncertainty ($\theta = 0.7 \rightarrow 0.3$, light blue) and low to high uncertainty ($\theta = 0.3 \rightarrow 0.7$, purple). Experts exhibited more pronounced context-dependent modulation of waiting time compared to novices. **(e)** Waiting times during the **first trial after a context switch**, summarizing data from (d). **(e₁)** Novice mice showed no significant difference between contexts. **(e₂)** Expert mice waited significantly longer in higher-uncertainty contexts ($p<0.05$, Wilcoxon signed-rank test). **(e₃)** Difference in modulation between novice and expert mice demonstrates increased context sensitivity with experience. Error bars and shaded regions represent standard error across mice.

change. We were particularly interested in determining whether mice infer the hidden context from input statistics or adapt their behavior based on rewards (or lack thereof) following their actions. We reasoned that if mice relied solely on reward feedback, we would only observe a significant behavioral change from the second trial after the context had changed (i.e., after experiencing the first success or failure in the new context). In contrast, if mice had already adapted their behavior in the first trial, it would suggest that they utilized sensory-based inference to inform their actions.

**First trial adaptation**  To investigate behavioral adaptation independent of feedback, we analyzed the mean waiting times in the first trials following each context change. Block switches occurred irregularly, preventing mice from anticipating context changes. In the first trial of a new context, mouse behavior already correlated more strongly with the steady-state waiting time of the new context than the old context's (Fig 2C). The significant change in behavior from the very first trial strongly indicates that the mice's policy adapts to implicit changes in environmental conditions through inference rather than solely relying on behavioral feedback.

To demonstrate that the observed changes in waiting time represent learned task-related behavior, we conducted an additional experiment comparing mice at different stages of training. We first trained a new group of 7 mice on the task in a single context ($\theta = 0.3$), allowing them to acquire basic familiarity with the environment. We then introduced these mice to the full task, which included unpredictable block switches between contexts with $\theta = 0.3$ and $\theta = 0.7$. This design allowed us to compare the performance of novice mice with that of mice with prolonged exposure to context switching.

Fig 2D illustrates the mean response times, in novice and expert mice, for the last four trials before a block switch and the first four trials after the switch. Fig 2E illustrates the mean waiting time for the first trial following a block switch, and the behavioral difference of the first trial between contexts. The results reveal a notable difference between the two groups. Compared to novice mice, expert mice demonstrated a significant change in waiting time from the very first trial of a new block.

This contrast between novice and expert performance indicates that the context adaptation we observe is indeed a learned task-related behavior. The ability of expert mice to rapidly adjust their waiting times in response to changes in context is not a simple response to auditory stimuli, nor is it merely a by-product of the increased frequency of *go* signals during high $\theta$ contexts. Instead, it represents a nontrivial cognitive skill acquired through experience with the task.

Our experiments with mice reveal three critical findings: (1) Mice adapt their behavior to the environment's latent variables; (2) this adaptation is primarily driven by inference from auditory input statistics rather than reward feedback; and (3) the ability to adapt from the first trial is a learned behavior, developed through experience with changing contexts. These results demonstrate mice are capable of flexible perceptual inference, allowing them to rapidly adapt to dynamic environments without relying solely on feedback. In the following sections, we will develop a Bayesian theory for optimal latent state inference in reinforcement learning settings and explore potential neural mechanisms underlying this capability using artificial neural networks.

## 3.2 Optimal decision-making in a dynamic environment with latent interactions

The environment introduced in the previous section requires the agent to simultaneously track a latent binary state and a latent contextual parameter that governs observation reliability. These dual uncertainties give rise to a challenging inference problem. To illustrate, consider a sequence of $k$ consecutive *go* signals $x_{t-k:t} = 1$. In an *unsafe* state, such a sequence should prompt the agent to update its estimate of the context parameter $\theta$, since repeated *go* observations in the unsafe state suggest a higher likelihood of observation flips. Conversely, in a *safe* state, the same stream of signals carries no information about $\theta$. Further complicating matters, a transition from unsafe to safe can occur mid-sequence, confounding the inference of $\theta$ and demanding a simultaneous estimation of both the state and the context.

This complexity reflects a form of *negative interaction information* [44,45] between the latent state and the context: each new observation not only strengthens their statistical coupling, but also amplifies the uncertainty surrounding each individual variable. While the state and the context may be independent *a priori*, they become statistically dependent once

conditioned on observations. Thus, resolving the joint inference of these hidden variables requires a unified treatment of their evolving likelihoods and the agent's actions.

Identifying an optimal policy in this setting goes beyond mere inference, as it requires the agent to select actions that maximize rewards in the face of dynamic uncertainty. We thus employ the framework of partially observed Markov decision processes (POMDPs) [46] to develop a rigorous Bayesian account of how an ideal observer should update its internal beliefs and choose actions. In the subsequent sections, we derive the belief-update equations for an optimal Bayesian observer and show how these updates drive a policy that achieves near-maximal reward. This analysis reveals how flexible perceptual inference can emerge through sequential belief-state updates that integrate observations with a learned internal model.

**Markov approximation** The task in Sect 2, designed to allow reliable mice behavior, contains two inherently non-Markovian features: (1) context changes occur only at trial boundaries, and (2) inter-trial intervals (ITIs) are sampled from a uniform distribution rather than an exponential distribution, which would satisfy the Markov property. To formulate the task as a partially observed Markov decision process (POMDP), we introduce two key approximations, both of which are well justified within the structure of our task.

First, we approximate context changes as a continuous-time Markov process. At each time step, the context can transition with a small probability $\epsilon$, selecting uniformly from $m$ discrete values $\theta' \in \{0, \frac{1}{m}, \frac{2}{m}, \dots, \frac{m-1}{m}\}$. The context transitions are described by:

$$P(\theta|\theta') = (1 - \epsilon)\delta_{\theta\theta'} + \frac{\epsilon}{m - 1}(1 - \delta_{\theta\theta'}), \tag{1}$$

where $\delta_{\theta\theta'}$ is the Kronecker delta function. This formulation approximates the discrete trial boundaries by allowing small, continuous changes in context over time. This approximation becomes exact as $\epsilon \to 0$ and seamlessly generalizes to continuous value contexts as $m \to \infty$. Thus, the choice of $\epsilon$ and $m$ can be adjusted to balance computational feasibility with task fidelity.

Second, we treat ITIs as part of the *unsafe* state. This simplification is justified because ITIs share similar input statistics with the unsafe state, and well-trained Bayesian agents typically avoid taking actions during ITIs. By merging these intervals with the unsafe state, we reduce the system to two latent states: *safe* ($s = 1$) and *unsafe* ($s = 0$). The state transitions are then governed by:

$$P(s|s') = \begin{bmatrix} 1 - \lambda & 0 \\ \lambda & 1 \end{bmatrix}, \tag{2}$$

where $\lambda$ is the probability of transitioning from the unsafe to the safe state.

By combining the independent transition functions in Eqs (2) and (1), the task can be approximated as a POMDP. In this framework, the agent observes only the binary input $x_t$, while the two underlying latent variables, state $s$ and context $\theta$, evolve according to a Markov process Fig 3A. This representation retains the key features of the task while simplifying its mathematical formulation, enabling us to derive the optimal policy using the POMDP framework Fig 3B.

**Belief states** In the framework of a POMDP, belief states provide a probabilistic representation of the agent's knowledge about the environment, encapsulating all relevant information from past actions and observations [47]. Specifically, a belief state is a probability distribution over the latent variables, effectively transforming the POMDP into a fully observable Markov decision process (MDP) over the belief space. This transformation enables decision-making based on a sufficient statistic of the agent's history, but solving the POMDP requires defining how beliefs are updated with new observations and how they guide optimal actions.

In our task, the belief state $\mathcal{P}_t(s, \theta|\boldsymbol{x}_{\leq t})$ represents the joint probability of the latent state $s_t$ and the context $\theta_t$ given the observation history $\boldsymbol{x}_{\leq t}$. **Although $s$ and $\theta$ are *a priori* independent, they become statistically dependent when conditioned on observations**. This conditional dependence arises because the same observations inform both the current state and the context, creating intertwined estimates.

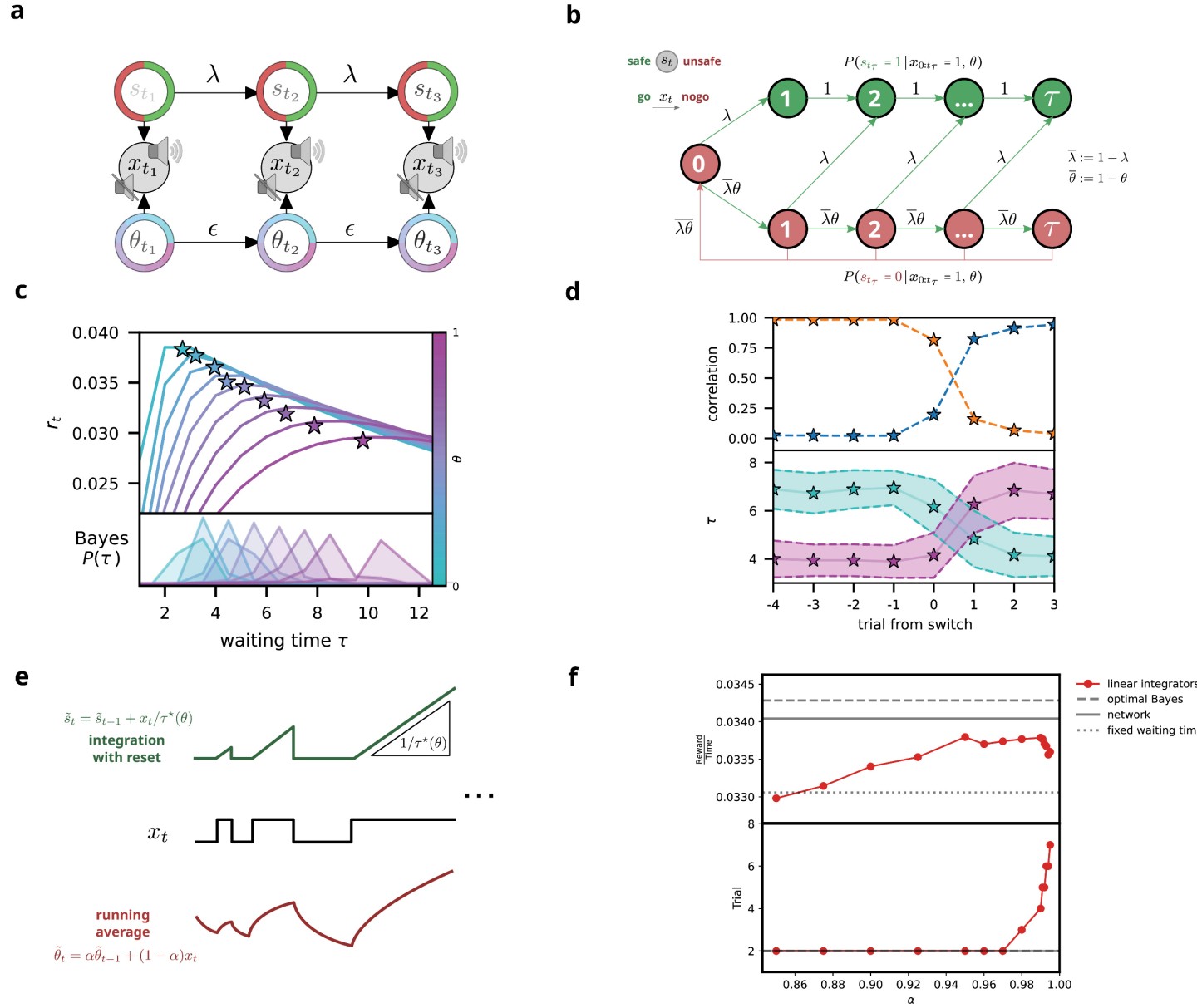

**Fig 3**. **Optimal Bayesian inference of latent state and context**. **(a)** Graphical model of the task structure. The latent state $s_t$ and context $\theta_t$ are independent sources of sensory observations $x_t$, transitioning at rates $\lambda$ and $\epsilon$, respectively. Conditioned on $x_t$, the state and context become dependent. **(b)** Markov chain describing the accumulation of $\tau$ consecutive *go* cues $x_t$ within a trial. A transition from *unsafe* (red) to *safe* (green) can occur at any time during a chain of *go* cues, with a *nogo* cue resetting $\tau$ to zero. This structure captures the probabilistic evolution of latent-state estimates based on sequential evidence. **(c)** *Top* Reward rate as a function of waiting time $\tau$ across contexts, illustrating optimal action timing. A Bayes-optimal agent (stars) that jointly infers latent state and context achieves near-maximal reward rates. *Bottom* Bayesian agent's waiting-time distributions across contexts; variance reflects uncertainty in current context estimates. **(d)** *Top* Correlation between trial waiting time and average waiting times from the previous (orange) and current (blue) blocks, demonstrating context-dependent first-trial adaptation (cf. Fig 2). *Bottom* Average waiting times for specific context transitions ($\theta_t = 0.3 \to 0.7$, purple; $\theta_t = 0.7 \to 0.3$, light blue). Shaded areas indicate Bayesian-posterior standard deviation. **(e)** Conventional drift–diffusion models. Integration with reset (*top*) and leaky integration (*bottom*) yield independent but biased estimates of $\hat{s}$ and $\hat{\theta}$. **(f)** Comparison of model performance with optimal Bayes policy, optimal fixed waiting time and example network. *Top* Reward rate. *Bottom* Earliest trial in which the model's waiting time became more strongly correlated with the current context than with the previous context.

To illustrate this dependency, consider a sequence of *go* signals $x_t = 1$. As the sequence grows, the probability of being in the safe state increases, but it also increases the likelihood of a high-uncertainty context (larger $\theta$). Resolving this interdependence requires maintaining and updating the full joint belief.

State and context estimators are derived from the posterior mean of the belief state:

$$\hat{s}_t = \langle s \rangle_{\mathcal{P}_t} = \sum_{s,\theta} s\, \mathcal{P}_t(s,\theta|\mathbf{x}_{\leq t}), \quad \hat{\theta}_t = \langle \theta \rangle_{\mathcal{P}_t} = \sum_{s,\theta} \theta\, \mathcal{P}_t(s,\theta|\mathbf{x}_{\leq t}), \tag{3}$$

where $\langle \cdot \rangle_{\mathcal{P}_t}$ denotes averaging with respect to the current belief $\mathcal{P}_t$. These estimators, $\hat{s}_t$ and $\hat{\theta}_t$, provide the agent's best estimates of the state and context at any given time. The estimate evolves by updating the joint belief as new observations arrive.

**Sequential updating of belief states** In dynamic environments, agents must continuously update their internal representations as new observations become available. In a Bayes-optimal setting, it is assumed that the agent has knowledge of the statistical structure of the environment, including the state transition probabilities $P(s|s')$, the context transition probabilities $P(\theta|\theta')$, and the observation likelihood $P(x_t|s,\theta)$. This knowledge could be acquired through prior experience or training. Using these known probabilities, the agent updates its belief state to optimally integrate new information with prior knowledge, ensuring that its internal belief reflects the current environment as accurately as possible.

We obtain a master equation for updating the belief state $\mathcal{P}_t(s,\theta|\mathbf{x}_{\leq t})$ using the Chapman-Kolmogorov equation [7,48]:

$$\mathcal{P}_t(s,\theta|\mathbf{x}_{\leq t}) = \mathcal{Z}^{-1}\, P(x_t|s,\theta) \sum_{s'} P(s|s') \sum_{\theta'} P(\theta|\theta')\, \mathcal{P}_{t-1}(s',\theta'|\mathbf{x}_{\leq t-1}), \tag{4}$$

where $\mathcal{Z}$ is a normalization factor ensuring that $\mathcal{P}_t(s,\theta|\mathbf{x}_{\leq t})$ sums to 1.

On the left-hand side of Eq (4), we have the current belief state. On the right-hand side, the update integrates the likelihood of the current observation $P(x_t|s,\theta)$, which reflects how well the current input aligns with the latent variables, the transition probabilities $P(s|s')$ and $P(\theta|\theta')$ which describe the dynamics of state and context, and the prior belief state $\mathcal{P}_{t-1}(s',\theta'|\mathbf{x}_{\leq t-1})$ which incorporates all information available up to the previous time step. Together, these components enable the belief state to evolve as new information becomes available, combining prior knowledge with real-time observations.

To initialize the belief state at the beginning of each trial, we assume that the world starts in the unsafe state ($s_t = 0$), leading to $\mathcal{P}_t(s=1,\theta) = 0$. The state in the previous step $s_{t-1}$ is known because the agent has just acted and received feedback. Consequently, the initial distribution of $\theta$ is given by the belief state at the end of the previous trial, conditioned on the known state at that time: $\mathcal{P}_t(s=1,\theta) = 0$ and $\mathcal{P}_t(s=0,\theta) = \mathcal{P}_{t-1}(\theta|s=s_{t-1})$. This setup allows the agent to leverage feedback from the environment in addition to sensory observations.

**Optimal policy** The optimal policy maximizes the expected reward rate, defined as the average reward over a fixed time horizon [10]. This requires balancing the need to gather sufficient evidence with the time cost of waiting, reducing the policy to determining the optimal number of consecutive *go* signals, $\tau$, required before acting, which depends on the context $\theta$.

For a given $\theta$, the probability of being in the safe state after $\tau$ consecutive *go* signals is

$$R(\tau;\theta) = 1 - \frac{b^\tau}{1 - c \sum_{k=0}^{\tau-1} b^k}, \tag{5}$$

where $b := (1-\lambda)\theta$ is the probability of observing a misleading *go* signal, and $c := (1-\lambda)(1-\theta)$ is the probability of observing a *nogo* signal.

The expected trial duration is

$$T(\tau; \theta) = \frac{\tau b^{\tau} + \sum_{k=0}^{\tau-1} b^k(\tau\lambda + c(k+1))}{1 - c\sum_{k=0}^{\tau-1} b^k}. \tag{6}$$

The reward rate is then given by

$$r(\tau; \theta) = \frac{R(\tau; \theta)}{T(\tau; \theta) + T_{ITI}}, \tag{7}$$

where $T_{ITI}$ is the average inter-trial interval. The optimal waiting time $\tau^{\star}(\theta)$ maximizes this reward rate

$$\tau^{\star}(\theta) = \arg\max_{\tau} r(\tau; \theta). \tag{8}$$

The curves in Fig 3C illustrate the relationship between $r(\tau; \theta)$ and $\tau$ for several values of $\theta$. The maximum of each curve indicates the optimal waiting time and the corresponding reward rate for that specific context.

In our task, the context $\theta$ is not directly observable and must be inferred from the input stream. The optimal waiting time, therefore, must account for this uncertainty by maximizing the expected reward rate, averaged over the belief state

$$\hat{\tau}^{\star} = \arg\max_{\tau} \langle r(\tau; \theta) \rangle_{\mathcal{P}_t}, \tag{9}$$

where the expectation is taken over the current belief $\mathcal{P}_t(\theta)$, reflecting the agent's estimate of the context based on all past observations.

A key property of the task is that, following a *nogo* signal, the probability of being in a safe state drops to zero. This intuitive result arises directly from the task structure and can be formally derived from the dynamics in Eq (4). Specifically, when a *nogo* signal is observed ($x_t = 0$), the likelihood of being in the safe state becomes zero, as $P(x_t = 0|s_t = 1, \theta) = 0$. Conversely, with each consecutive *go* signal, the probability of being in the safe state $\hat{s}_t = \langle s \rangle_{\mathcal{P}_t}$ increases monotonically.

Using the optimal waiting time $\hat{\tau}^{\star}$, we calculate the belief threshold $\hat{s}^{\star}$ at which the agent should act. This threshold represents the minimum belief in the safe state required to maximize the reward rate:

$$\hat{s}^{\star} = \langle P(s = 1|\theta, \bar{\mathbf{x}}_{\hat{\tau}^{\star}}) \rangle_{\mathcal{P}_t}, \tag{10}$$

where $\bar{\mathbf{x}}_{\hat{\tau}^{\star}} = (0, 1, 1, 1, ...)$ denotes a sequence starting with a *nogo* signal $x_t = 0$, followed by $\hat{\tau}^{\star}$ consecutive *go* signals $x_t = 1$.

This belief threshold $\hat{s}^{\star}$ defines the critical confidence level at which the agent should act. It captures the trade-off between collecting more evidence and the time penalty incurred by waiting too long. As the waiting time exceeds $\hat{\tau}^{\star}$, the expected reward rate decreases monotonically. Thus, the agent's policy is to act whenever the current belief in the safe state $\hat{s}_t$ reaches or exceeds the threshold $\hat{s}^{\star}$ [49].

Importantly, $\hat{s}^{\star}$ varies with the belief $\mathcal{P}_t$, and in particular with the estimated context $\hat{\theta}$. In more uncertain environments (higher $\hat{\theta}$), the threshold $\hat{s}^{\star}$ decreases, meaning the agent requires less certainty to act. This adjustment reflects the need to act more readily in contexts with ambiguous input to avoid excessive delays.

The belief threshold enables the formulation of a deterministic policy based solely on the current belief state

$$\pi^{\star}(a_t|\mathcal{P}_t) = \begin{cases} 1 \text{ (act)}, & \delta(\mathcal{P}_t) \geq 0, \\ 0 \text{ (don't act)}, & \text{otherwise}, \end{cases} \tag{11}$$

where we introduced the *decision variable* $\delta(\mathcal{P}_t) = \hat{s}_t - \hat{s}^\star$. This policy directs the agent to act when its belief in the safe state exceeds the optimal threshold, translating the agent's understanding of state and context into a concrete action rule.

Together, the belief-update rule in Eq (4) and the policy in Eq (11) define a Bayes-optimal strategy for decision-making in dynamic environments. By inferring latent states and contextual reliability from a single observation stream, this framework integrates evidence in real time, resolves input ambiguity, and adapts to changing statistics; it provides a theoretical benchmark for understanding flexible perceptual inference.

### 3.3 Dynamics of Bayes-optimal perceptual decision-making in dynamic environments

Having derived a Bayes-optimal solution to account for the joint inference of state and context (Sect 3.2), we now apply this theoretical framework to the change-detection task introduced in Sect 2. The Bayesian solution allows us to quantify how an ideal observer integrates ambiguous observations and adapts to implicit context changes. Crucially, it also allows us to study how an ideal drift-diffusion model would implement simultaneous inference of the latent state and context.

**Performance of a Bayes-optimal actor.** We first assessed the performance of an agent that updates its belief states according to Eq (4) and follows the optimal policy in Eq (11). As illustrated in Fig 3C (top), the Bayes-optimal actor attains near-perfect performance, closely approximating an agent with explicit knowledge of the context. The modest performance gap arises from residual uncertainty in the estimated context, which broadens the distribution of waiting times $P(\tau)$ (bottom of Fig 3C). This efficiency gap reflects the central challenge of *flexible perceptual inference*: using noisy and partial observations to infer, online, both the binary state and the reliability of incoming signals.

**Adapting to a new context within the first trial.** One of the hallmarks of flexible perceptual inference, as laid out in the introduction, is rapid adaptation to changing environments without relying on trial-and-error. Our Bayesian framework meets this requirement by continuously updating the context estimate from each incoming observation. Fig 3D demonstrates how the Bayes-optimal actor adapts within the very first trial of a new context block: correlations between waiting times in the current trial and in previous blocks reveal immediate behavioral shifts, even before any explicit feedback. This ability to adapt in real-time matches the behavior observed in mice (Fig 2C and 2D), suggesting that animals may employ similarly sophisticated inference when faced with dynamic uncertainties.

**Tractable approximations**  The Bayes-optimal solution derived above represents the theoretical upper bound on performance for any agent that must infer both state and context from ambiguous observations. By maintaining joint probability distributions over both latent variables, the Bayesian agent achieves near-maximal reward rates, approaching but never quite matching the performance of a context-aware agent (Fig 3C). This gap reflects the fundamental cost of uncertainty. However, even this near-optimal performance demands computational resources that may exceed the capacity of biological systems [50,51].

Bayesian computations face severe scalability challenges in neural implementation. The curse of dimensionality becomes prohibitive for continuous contexts: maintaining probability distributions over all possible $\theta$ values requires exponentially scaling resources. Even in our discrete setting, the joint belief state $\mathcal{P}_t(s, \theta | \boldsymbol{x}_{\leq t})$ tracks $2m$ values per time step. For biologically realistic scenarios, exact Bayesian inference becomes computationally intractable.

Could simpler mechanisms achieve comparable performance? The structure of the optimal solution offers a compelling insight. Given perfect knowledge of context $\theta$, the optimal strategy reduces to waiting exactly $\tau^*(\theta)$ consecutive *go* signals before acting (Eq (8)). This waiting time perfectly balances evidence reliability against the speed-accuracy tradeoff. Crucially, a simple integrator can implement this strategy [10,52]: accumulate evidence at rate $1/\tau^*(\theta)$, reset to zero on each *nogo* signal, and trigger action when the threshold of one is reached. Such a mechanism would match the performance peaks in Fig 3C—if only the context was known.

The crux then lies in estimating the context from the observation stream itself. A natural heuristic tracks the running average of *go* signals through the recursive update $\tilde{\theta}_t = \alpha \tilde{\theta}_{t-1} + (1 - \alpha)x_t$, where $\alpha$ controls the integration timescale [53].

This approach replaces the joint inference with two linear integrators (Fig 3E). While computationally tractable and biologically plausible, this simplified model suffers from fundamental limitations.

First, the choice of $\alpha$ imposes an inescapable tradeoff. Short integration windows ($\alpha$ small) yield high-variance estimates but rapid adaptation to context changes. Long windows ($\alpha$ large) provide stable estimates but sluggish responses to environmental changes. No single value can optimize both precision and flexibility [15,17,54].

Second, and more fundamentally, this heuristic systematically overestimates the true context ($\tilde{\theta}_t \geq \theta_t$). The running average cannot distinguish between misleading *go* signals in the unsafe state—which should contribute to the context estimate—and reliable *go* signals in the safe state—which should not. It is possible to compensate by adjusting the mapping from $\tilde{\theta}$ to $\tau^*$. However, the overestimation bias varies nonlinearly with the true context, excluding simple corrections.

These limitations highlight a deeper challenge: even simplified heuristics require sophisticated neural implementation. Converting a context estimate into either a dynamic integration rate or an adaptive threshold demands computational machinery that may itself approach the complexity of the original problem. How neural networks discover and implement such mechanisms through learning alone remains unclear. In the following sections, we investigate this question directly by training recurrent neural networks on our task and analyzing the computational strategies that emerge.

### 3.4 Neural mechanisms of approximate Bayesian computation

Having derived the Bayes-optimal solution and examined its computational demands, we now investigate how neural networks solve this task. The joint inference of state and context requires maintaining probability distributions over multiple latent variables—a computation that may exceed biological constraints. Can neural circuits discover efficient approximations through learning alone? To address this question, we trained recurrent neural networks using reinforcement learning, providing only reward signals without explicit instruction for probabilistic computation. By analyzing both behavioral performance and internal representations, we examined whether and how these networks implement the principles of flexible perceptual inference.

**Network architecture and training**  To solve our task, we used a deep reinforcement learning approach, implementing an actor-critic framework [55] with 100 long short-term memory (LSTM) units [56] (Fig 4A). This architecture was chosen for its ability to handle sequential input in decision-making tasks and learn from temporal correlations in the input data. Although LSTM networks do not directly correspond to biological neurons, they allow us to study a neuronal implementation of the inference dynamics. Furthermore, LSTM networks have been suggested as models for the prefrontal cortex [57] and can also be implemented using microscopic neuronal circuits [58].

The critic subnetwork is designed to estimate the value function, predicting the current expected future reward. The output of the actor subnetwork, denoted $Q_t$, parametrizes the probability of acting at each step, which is given by $P_{train}(a_t = 1) = (1 + \tanh Q_t)/2$ during the training phase and $P_{test}(a_t = 1) = (1 + \mathrm{sign}\, Q_t)/2$ during the testing. The sign function represents the deterministic threshold expected from our theory, while the soft threshold allows necessary exploration during training.

The network receives three inputs at each time step: the current observation, the action taken in the previous step, and the reward received in the previous step. This input structure allows the network to learn from the observations and previous actions and rewards. We trained the network using a reinforcement learning framework, specifically applying it to the task described in Sect 2. Our training procedure used a Monte Carlo approach, which involves learning from complete episodes rather than after each time step or behavioral outcome [55]. After every 20 trials, each consisting of 30 steps on average, we update the network weights using backpropagation through time [59].

**Network outperform integrators model**  Our analysis demonstrates that trained RNNs closely approximate the performance and behavior of an optimal Bayesian agent across various context values (Fig 4B, top panel). The RNNs achieve near-optimal performance, exhibiting qualitatively similar distributions of waiting times (Fig 4B, bottom panels) compared to the Bayesian agents. Furthermore, networks display rapid context adaptation within the first test after a block

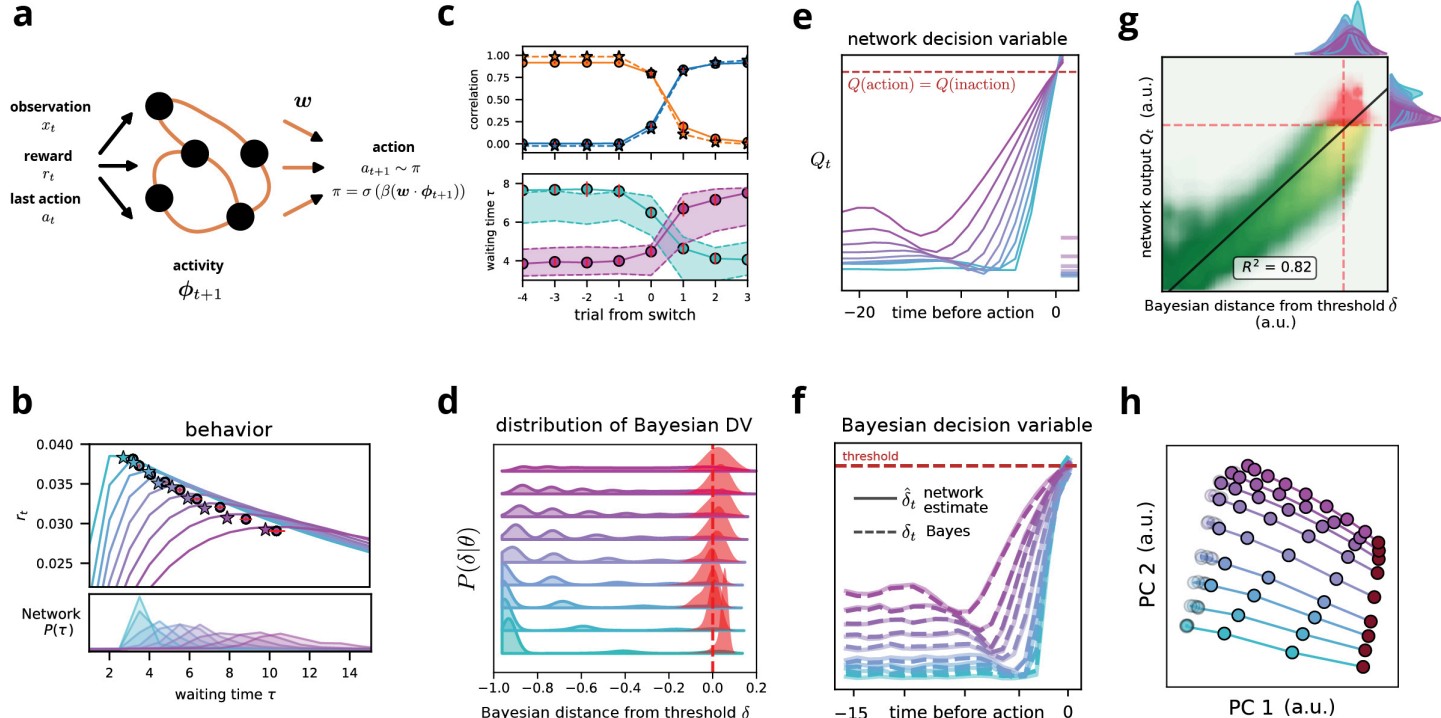

**Fig 4**. **Neural networks trained via reinforcement learning approximate Bayesian inference through learned recurrent dynamics. (a)** Actor-critic LSTM architecture (100 units). The network receives observations $x_t$, previous actions, and rewards, outputting the decision variable $Q_t$. **(b)** Networks (circles, $n = 10$) achieve near-Bayesian performance (stars) across contexts. *Top* Reward rates. *Bottom* Waiting time distributions. **(c)** First-trial adaptation following context switches. *Top* Waiting times correlate with new (blue) rather than previous (orange) context. *Bottom* Immediate adjustment for transitions between $\theta = 0.3$ and $0.7$. **(d)** Bayesian decision variable $\delta_t$ distributions (colored by context) converge to consistent threshold at action (red). **(e)** Context-dependent ramping of network output $Q_t$. Horizontal lines: Initial values vary with context. **(f)** Decoded decision variable (solid) tracks theoretical Bayesian $\delta_t$ (dashed). **(g)** Network output strongly correlates with Bayesian decision variable ($R^2 = 0.82$; Fig 4E–4G). Color: time since *nogo*. **(h)** PCA reveals low-dimensional dynamics with PC1 encoding decisions (64% variance) and PC2 encoding context (13.4% variance). Trajectories separate by context $\theta$. Error bars: mean ± s.d. ($n = 10$ networks).

switch (Fig 4C). Across all contexts, the network outperforms the disjoint linear integrator model, nearly matching the behavior of an optimal Bayesian actor in both the average reward rate and in adaptation to context switch (Fig 3F).

**Single trial analysis.** Near-optimal average performance does not prove that networks implement Bayesian inference—multiple mechanisms could yield similar reward rates. To test whether networks truly approximate Bayesian computation, we must examine their trial-by-trial decision dynamics. This presents a methodological challenge: We cannot directly compare the behavior of neural and Bayesian agents in parallel, as their actions actively shape the environment state. The independent operation of both agents would lead to divergent input histories, limiting us to comparing only their average performance.

We resolved this by computing the Bayesian decision variable $\delta(\mathcal{P}_t)$ along trajectories controlled by the network. If the network implements Bayesian-like inference, it should act when $\delta(\mathcal{P}_t)$ each is the optimal threshold. Indeed, at the moment of network action, the Bayesian decision variable consistently clusters near its theoretical threshold across all contexts (Fig 4D), while showing broad distributions during ongoing trials.

This threshold alignment extends beyond action timing. Throughout entire trials, the network output $Q_t$ maintains a strong linear relationship with the Bayesian decision variable ($R^2 = 0.82$; Fig 4E–4G). This correlation persists in different

stages of evidence accumulation, from early uncertainty to decision commitment, demonstrating that the network continuously tracks the optimal belief state rather than simply learning appropriate action times. Such faithful tracking of the Bayesian computation throughout the decision process explains the network's near-optimal performance and rapid adaptation to context changes.

**Neural dynamics** Having established that the network's output tracks the Bayesian decision variable, we next investigated the neural mechanisms underlying this computation. If the network merely implements linear integration with adaptive thresholds, its internal dynamics should reflect simple accumulation. Instead, we find that the network has discovered a fundamentally different computational strategy.

Principal component analysis reveals that network activity unfolds in a low-dimensional manifold: three principal components capture 88% of the variance (Fig 4H). This dimensionality reduction exposes the network's computational architecture. The first PC (64% variance) encodes the decision variable itself, correlating strongly with both the network output $Q_t$ and the Bayesian $\delta_t$ ($R^2 = 0.91$). The second PC (13.4% variance) represents the context estimate ($R^2 = 0.87$ with $\theta$), while the third PC (10.6% variance) coordinates post-action reset dynamics, ensuring proper initialization for subsequent trials

The critical insight emerges from examining how these components evolve during evidence accumulation. We compared three dynamical systems: the Bayesian estimators (Fig 5A), a heuristic model with two disjoint linear integrators (Fig 5B), and the network's learned representations (Fig 5C). While the network dynamics broadly resemble the Bayesian solution, the specific mechanisms reveal fundamental differences from linear integration.

Consider first how each system processes consecutive *go* signals. A linear integrator accumulates evidence indiscriminately—each *go* signal simultaneously drives action and increases the context estimate (Fig 5B). The network's context representation (PC2) violates this prediction dramatically: after an initial transient, PC2 remains nearly constant during *go* sequences (Fig 5C). This selective gating mirrors the Bayesian solution, which recognizes that *go* signals in the safe state provide no information about context reliability. The network has learned to suppress context updates precisely when such updates would be misleading.

The processing of *nogo* signals reveals an even sharper distinction. While both models reset their state estimate upon receiving *nogo*, they differ fundamentally in context updating. The linear integrator simply decreases its context estimate, treating *nogo* as negative evidence (Fig 5B). The Bayesian model implements a counterintuitive but optimal strategy: *nogo* signals trigger sharp *increases* in context estimation (Fig 5A). This occurs because *nogo* confirms the unsafe state with certainty, implying that all preceding *go* signals were misleading, evidence of a greater uncertainty of the context. Remarkably, the network's context representation (PC2) exhibits this same non-monotonic update pattern (Fig 5C) as the Bayesian actor. Notably, the networks cannot implement a discontinuous jump, and the sharp reset is implemented in a few steps. This numerical discrepancy disappears when the internal dynamics of the network becomes continuous and is limited only by the internal time constant of the network.

To quantify how faithfully the network implements these computations, we tested whether fixed linear decoders could extract the relevant variables from network activity. Using cross-validated regression (10-fold, 80-20 split), we reconstructed the Bayesian decision variable with remarkable fidelity ($R^2 = 0.993$; Fig 4H). For comparison, decoders also accurately recovered the ground-truth state (accuracy = 0.90) and context ($R^2 = 0.92$) but with lower accuracy. Critically, attempting to decode a linear integrator's output yielded yet lower accuracy ($R^2 = 0.84$), confirming that the network implements Bayesian-like rather than linear dynamics.

Our analyses reveal that neural networks trained via reinforcement learning discover an efficient approximation to Bayesian inference without explicit instruction. Rather than implementing full probability distributions, the network learns to embed the essential nonlinear dynamics—selective evidence gating, context-dependent resets, and joint state-context updates—within its recurrent connectivity. This emergence of principled computation from reward signals alone demonstrates how reinforcement learning can give rise to flexible perceptual inference.

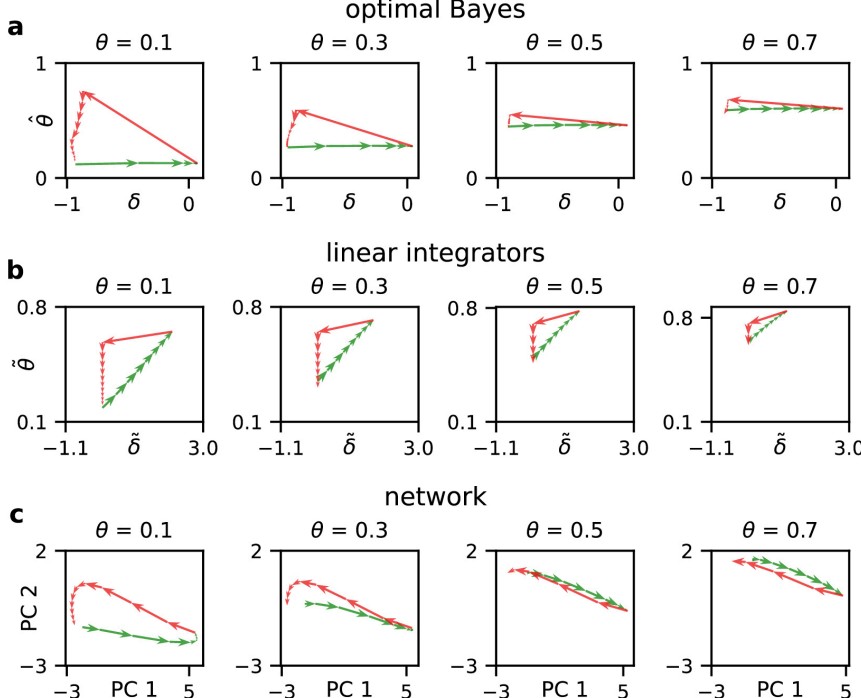

**Fig 5. Comparative dynamical analysis of evidence accumulation in the normative models and networks**. **(a)** Bayes-optimal dynamics. Trajectories show the evolution of state ($\hat{s}_t$) and context ($\hat{\theta}_t$) estimators derived from the belief state Eq (4) during sequences of 8 consecutive *go* signals (green arrows) followed by *nogo* signals (red arrows). Note the nonlinear context updates: *nogo* signals paradoxically increase $\hat{\theta}_t$ as they confirm preceding *go* signals were misleading, whereas *go* signals in the safe state (high $\hat{s}_t$) produce minimal context updates. **(b)** Linear integrator dynamics. The decision variable $\tilde{\delta}_t$ and context estimate $\tilde{\theta}_t$ evolve via leaky integration ($\alpha = 0.9$; see Sect 3.3). Both variables update monotonically with each observation, lacking the state-dependent updates observed in the joint belief update of Bayesian inference. **(c)** Example network dynamics. Neural trajectories projected onto the first two principal components (PCs) reveal Bayesian-like computation. PC1 (64% variance) encodes the decision variable ($R^2 = 0.91$ with $\delta_t$), while PC2 (13.4% variance) represents context ($R^2 = 0.87$ with $\theta_t$). The network implements selective gating: PC2 remains nearly constant during *go* sequences, but increases following *nogo* signals, mirroring the Bayesian solution rather than linear integration. Each column shows the update dynamics for exact same conditions under the three model with different context values ($\theta = 0.1, 0.3, 0.5, 0.7$).

## 3.5 Approaching Bayes-optimal performance requires training the recurrent connections

A recent study showed that optimal Bayesian estimators can emerge from neural networks trained for value prediction [60]. In particular, estimators over beliefs can be linearly decoded even from untrained networks, suggesting that the intrinsic dynamics of random networks might be sufficiently rich to support Bayesian computation. In this case, Bayesian inference can be learned through reservoir computing [61], training only the readout weights.

However, there are key differences between value prediction and perceptual decision-making. In particular, we consider an *operant* task that demands discrete decisions. The key distinction lies in the behavioral output. Whereas value prediction tasks involve estimating a continuous variable, our task requires categorical choice (act or wait) based on a dynamic decision variable. This difference is crucial, as optimal performance in our task depends on accurate single-trial state estimation. The average network output, which might suffice for value prediction, may not capture the temporal precision required for adaptive behavior in dynamic environments.

To examine whether random networks can maintain belief-like representations in this more challenging task setting, we used a "naive" model. This model consists of an RNN with fixed random weights for both recurrent and input connections. As in reservoir computing, only readout weights were trained, which feed into a *softmax* layer to produce action probabilities (Fig 6D).

We found that retaining the *softmax*-based probabilistic action selection was essential for the functionality of the naive model. This differs from the fully trained network, which could learn a deterministic policy as prescribed by the theory (using a hard threshold in the final layer). However, the naive model required stochastic action selection to perform acceptably.

The performance of this random RNN model was poor (Fig 6A). Furthermore, it exhibits a different qualitative behavior. While optimal performance requires longer waiting times at higher $\theta$ values, the random RNN model showed the opposite pattern, with decreased waiting times at higher $\theta$ values (Fig 6B). This maladaptive behavior is a direct result of stimulus statistics. Without proper context inference, the network responds to go cues by simply increasing its action probability, ignoring the broader temporal structure of the task.

Despite the poor behavioral performance, we could decode Bayesian state and context estimators from the network's activity with relatively high fidelity. Using the same cross-validation methods applied to trained networks, we found that random dynamics had strong predictive power, albeit slightly lower than fully trained networks ($p(\text{correct}) = 0.89 \pm 0.001$ for the state and $R^2 = 0.91 \pm 0.003$ for the context estimators; mean $\pm$std). These results align with the findings of [60], which successfully decoded Bayesian estimators from random networks in a value prediction task. However, they raise the question of why the network's performance still is so poor.

**Untrained networks fail when it matters the most.** The apparent contradiction between the successful decoding of Bayesian estimators and poor behavioral performance (Fig 6G left) in random RNN models stems from a specific limitation in their representation of latent variables. While network activity allows good decoding on average, it fails critically at the most behaviorally relevant moment, after several consecutive *go* signals, when the agent needs to make a decision. Specifically, when the decision variable approaches the action threshold, the random network output $Q_t$ shows no meaningful differentiation between contexts or the number of consecutive *go* signals (Fig 6H left).

Random recurrent networks have inherent limitations in their temporal integration capabilities. Classical random networks exhibit exponentially decaying memory traces [62], while random LSTM architectures can maintain somewhat longer dependencies [63]. This memory constraint allows linear decoders to differentiate contexts based on recent history, but the discrimination inevitably fails after several consecutive go signals as the memory trace degrades. Assuming a random network, the dynamics of the output $Q_t$ can be captured by a simple linear integration model

$$z_t = \alpha z_{t-1} + (1 - \alpha)x_t \tag{12}$$

$$Q_t = \beta z_t + \epsilon_\theta, \tag{13}$$

where $z_t$ represents a leaky linear integration of the observations $x_t$ and $\alpha$ controls the memory trace. The output $Q_t$ is a linear transformation of the integrator state with gain $\beta$ and baseline $\epsilon$. This minimal model accounts for $99.0 \pm 0.2\%$ of the single-trial variance while explaining only $74.0 \pm 1.6\%$ of the variance in fully trained networks, confirming that random networks implement a fundamentally simpler computational strategy.

**Learning the recurrent weights is required even when explicit context is provided.** The failure of random networks to maintain context-dependent representations near decision boundaries raises a fundamental question: Does this limitation stem from an inability to infer context or from an inability to implement proper context-dependent integration even with perfect context knowledge? To differentiate between these possibilities, we modified our random RNN model by directly supplying the context as an additional input (Fig 6E). This approach isolates the role of context inference from the computational demands of context-dependent evidence accumulation.

With explicit context information, the random network model demonstrated a significant qualitative improvement in its behavioral strategy. In particular, the mean waiting time increases with $\theta$, as expected from optimal behavior and observed in mouse data (Fig 6B). This increase is deterministic for the trained actors (Fig 6C). This qualitative change

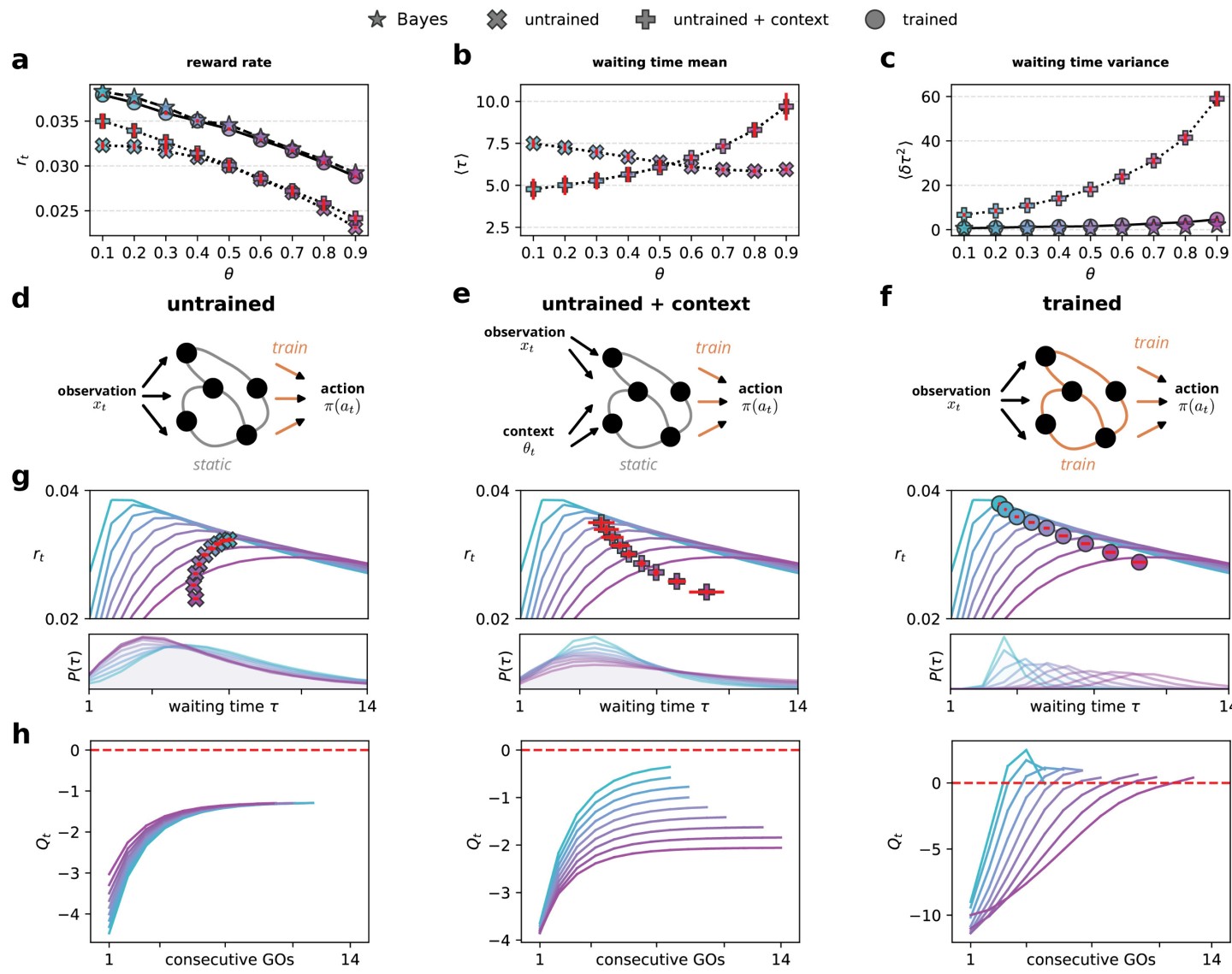

**Fig 6. Learning recurrent dynamics is necessary for implementing sequential inference of both state and context**. (a) Reward rate as a function of the context parameter $\theta$ for different network models (symbols indicated in the legend). Only trained networks achieve Bayesian-level performance. (b) Mean waiting time $\langle\tau\rangle$ as a function of $\theta$ for untrained networks (×) and untrained + context networks (+). Adding explicit external context improves the ability of a linear readout to modulate waiting times, increasing them for higher $\theta$. (c) Variance of waiting times $\langle\delta\tau^2\rangle$ as a function of $\theta$ across models. Both the Bayes-optimal agent and the trained and untrained + context networks show reduced variability compared to the untrained model, whose high trial-to-trial variance limits performance. (d–f) Network schematics for each model. (d) Untrained networks receive raw observations $x_t$ but lack learned dynamics. (e) Untrained + context networks receive explicit context $\theta_t$ alongside observations. (f) Trained networks learn recurrent dynamics that approximate the sequential Bayesian decision process. (g) *Top* Relationship between waiting times $\tau$ and average reward rate $r_t$ across models, compared to the context-aware Bayesian reward curve (see Figs 3 and 4). *Bottom* Waiting-time distributions across models, showing the effect of learning on action timing. (h) Evolution of the decision variable $Q_t$ as a function of the number of consecutive *go* signals across contexts. *Left* Untrained networks fail to distinguish contexts, relying on stochastic behavior. *Center* Explicit context improves separation of decision dynamics. *Right* Trained networks implement an optimal deterministic policy, rapidly adapting $Q_t$ to context.

suggests that access to context information enables the network to implement a basic form of context-dependent integration. Still, quantitative analysis revealed persistent suboptimal performance (Fig 6G middle), particularly pronounced at

high $\theta$ values where reward rates remained significantly below those achieved by the Bayesian model and fully trained networks (Fig 6A).

The random RNN model shows decreased performance despite achieving appropriate mean waiting times due to excessive response variability (Fig 6C). This limitation comes from the fixed memory capacity of the network, which restricts its ability to integrate *go* cues beyond a certain duration reliably. To compensate for limited memory and achieve longer mean waiting times, the network implements a suboptimal strategy: maintaining the output $Q_t$ at a context-dependent distance from the stochastic threshold, effectively trading off precision for approximate temporal control. This bias-variance trade-off allows the network to increase average waiting times through reduced action probabilities but at the cost of high trial-to-trial variability.

Similarly to the random network that did not receive an explicit context input, this mode can be explained by the linear integrator model $z_t$ shown in equation Eq (12), where the readout is given by $Q_t = \beta z_t + \epsilon_\theta$. Here, the baseline depends on the explicit context through the set of parameters $\epsilon_\theta$. As before, the linear model explains the output trajectory of the random model (with explicit context) very well (explained variance $99.1 \pm 0.6\%$) while failing to explain fully trained networks ($74.0 \pm 1.6\%$) (Fig 6H).

This systematic analysis of random networks reveals fundamental limitations in implementing Bayes-optimal behavior, even when provided with perfect context information. The now necessary reliance on stochastic policies starkly contrasts the deterministic Bayesian solution that uses internal belief states. Linear modeling exposed specific computational deficits: random networks do not modulate their integration rate or decision threshold. When context is provided, random networks improve their behavior by making crude threshold adjustments. In contrast, trained recurrent networks (Fig 6F) develop internal dynamics that independently control both integration rate and baseline in the precise and complementary manner predicted by Bayesian theory. These results demonstrate that optimal inference in our task emerges specifically through the training of recurrent connectivity, supporting a mechanistic link between reinforcement learning and flexible perceptual inference.

## 4 Discussion

We investigated how biological and artificial agents solve a fundamental challenge in perception: simultaneously inferring what is happening and how reliable their sensory information is. Using a change-detection task where both a latent state and its observational context must be inferred from ambiguous inputs, we uncovered the computational principles underlying *flexible perceptual inference*—the ability to adapt evidence integration in real-time without feedback or reward.

Mice exhibited remarkable cognitive flexibility, adjusting their waiting times within the first trial following context changes (Fig 2C and 2D), before receiving any reward feedback. This immediate adaptation demonstrates that mice perform statistical inference on sensory inputs rather than relying on trial-and-error learning when adapting to change of context in our task. Our theoretical analysis revealed why this problem is computationally challenging: the latent state and context, though *a priori* independent, become statistically coupled when conditioned on observations, necessitating joint inference that simple linear integrators cannot achieve (Sect 3.2).

Remarkably, recurrent neural networks trained by reinforcement learning discovered near-optimal solutions without explicit instruction in probabilistic computation (Fig 4). These networks learned a heuristic solution that implements key nonlinear features of Bayesian inference, including selective evidence gating and state-dependent updates. While the full Bayesian computation is likely intractable by neural systems, analysis of the evidence accumulation dynamics reveals key features that allow better heuristics. Critically, these capabilities emerged specifically through the learned recurrent connectivity; networks with fixed random weights failed even when provided perfect context information (Fig 6). This establishes flexible perceptual inference as a learnable capability that requires coordinated recurrent dynamics, not merely readout adjustments.

**Inference-based learning and cognitive flexibility.** Our findings strengthen the growing evidence for latent state inference in animals [64–66], providing a concrete neural example of this capability. We interpret dynamic adjustments in evidence integration rates and decision bounds as hallmarks of an adaptive internal inference model that supports flexible perceptual inference. This ability to update internal representations based on sensory input statistics, rather than relying solely on reward feedback, suggests a form of inference that transcends simple stimulus-response associations or purely reward-driven learning. From a computational perspective, our work extends the framework of meta-learning [55]—the concept of *learning to learn*—to encompass *learning to infer*. By demonstrating that recurrent neural networks can acquire near-optimal Bayesian inference through reinforcement learning, we propose a plausible neural mechanism underlying the cognitive flexibility exhibited by animals in dynamic environments.

**Neural signatures of flexible perceptual inference.** Our work bridges a critical gap between Bayesian theory and neural implementation. While animals demonstrably integrate uncertainty into decisions [2,67,68], the neural mechanisms underlying Bayesian computation remain elusive [69,70], particularly when exact inference becomes intractable due to high dimensionality or continuous variables. Our trained networks offer a solution to this puzzle: they reveal how neural circuits can implement efficient approximations to Bayesian inference through learned recurrent dynamics.

Specifically, our analysis identifies testable neural signatures of flexible perceptual inference. Neural populations performing this computation should exhibit: (1) low-dimensional dynamics with separable representations of state and context (Fig 4G), (2) nonlinear update rules where identical inputs produce context-dependent responses (Fig 5), and (3) selective gating that prevents context updates during certain states. These predictions are particularly relevant for brain regions implicated in adaptive behavior and state inference, such as orbitofrontal cortex [65] or posterior parietal cortex during evidence accumulation tasks. Future experiments could test whether neural populations in these regions exhibit the characteristic dynamics we identified, providing a direct link between our computational framework and biological implementation.

**Recurrent neural networks as models for complex cognitive processes.** Training RNNs to develop task-specific dynamics is essential for neural modeling of cognitive processes in complex, uncertain environments. We found that relying solely on readout weights from random RNNs failed to capture the intricate computations required for our task, consistent with the limitations of reservoir computing in temporally dependent tasks [61]. In contrast, training the recurrent connectivity enabled the networks to perform probabilistic computations in partially observable environments, where both latent states and contextual information must be inferred simultaneously. These results extend previous work demonstrating the role of recurrent dynamics in probabilistic computations [71–73] to more complex cognitive tasks. By training RNNs on such tasks, we uncover neural mechanisms underlying flexible perceptual inference. This approach bridges theoretical models and experimental neuroscience, offering a powerful framework for studying the mechanisms underlying complex cognitive functions. However, we note that a feedforward network can likely solve the task similarly if provided in a long enough window of recent activity at its input. This is because the Bayesian heuristic can be framed as a nonlinear yet deterministic operation on the observation history.

**Importance of operant tasks in studying decision-making.** Our findings highlight the critical role of operant tasks in studying decision-making, as they reveal behaviors that cannot be captured by state estimation or reward prediction alone. While latent variables and their Bayesian expectations can be decoded even from random networks [60], such networks exhibit poor task performance. This demonstrates that merely reading out mean or typical values from a network, as often done in reinforcement learning studies, is insufficient to assess its computational principles. By examining single-trial behavior in operant tasks, we combine normative principles with network simulations, revealing the subtleties of learned neural representations. This approach highlights the need for caution when interpreting neural data based solely on decoding specific variables, as it may obscure the underlying computational principles [74].

In summary, our results establish that the rapid trial-by-trial adaptation seen in both mice and artificial neural networks arises from a unifying principle of *flexible perceptual inference*, whereby latent states and contexts are jointly inferred. By

demonstrating how recurrent connectivity can naturally implement Bayesian-like computations under reinforcement learning, we highlight a plausible computational basis for real-time cognitive flexibility. Our results offer testable predictions for neural circuits engaged in adaptive decision-making, for example, in the orbitofrontal cortex. Future work should aim to identify how such inference is implemented at the circuit level and how it interacts with reinforcement learning to shape behavior in dynamic environments.

## Methods

### Ethics statement

All protocols and procedures were approved by the Institutional Animal Care and Use Committees at The Hebrew University of Jerusalem and were in accordance with the National Institutes of Health Guide for the Care and Use of Laboratory Animals.

### Animals

All experiments were conducted using male C57BL/6 mice under water restriction (age: 2-4 months, weight: 20-30 g). Mouse weight was monitored daily and extra water was provided if needed to ensure that mice maintain no less than 80% of their original weight. Mice were housed in a 12-hour light/dark cycle with ad libitum access to food. All procedures were approved by the Institutional Animal Care and Use Committee (IACUC) of Hebrew University.

### Surgical procedures

To implant head bars for head fixation, mice were anesthetized using isoflurane (SomnoSuite, Kent Scientific USA) and placed in a stereotaxic apparatus (KOPF 962, David Kopf instruments, USA). Meloxicam (2 mg/kg) was injected subcutaneously for systemic analgesia and 0.1 ml lidocaine (2% solution) was injected subcutaneously before incising the scalp and exposing the skull. A 20 mm long and 3 mm wide head-post was then cemented to the skull horizontally using dental cement (C&B Super-bond dental cement, Sun Medical, Japan) and secured using dental resin (Pi-ku-plast, Bredent, Germany). Post-operative analgesia was administered and mice were allowed to recover for 2 to 3 days before beginning a 1-week water restriction schedule prior to training.

### Animal task implementation

Head-fixed mice were trained to perform an auditory change detection task. Each trial was discretized into 0.2-second time bins, during which the trial remained in one of four latent states: *unsafe*, *safe*, *premature*, or inter-trial interval (*ITI*). In all states, except the safe state, beeps (i.e., go cues) were generated with probability $\theta$ (meaning that if $\theta = 0.3$, a beep occurred in 30% of bins). In the *safe* state, beeps occurred with probability 1 (i.e., in every time bin).

Each trial began in the *unsafe* state. If the mouse withheld licking, the state transitioned from *unsafe* to *safe* after a delay of $0.8 + 0.2|x|$ seconds (where $x \sim \exp(\lambda = 0.1)$ and $|x|$ is the floor of $x$). Licking prematurely during the *unsafe* state triggered the *premature* state, which lasted for 7 seconds, before the state transitioned to the *ITI*. In the *safe* state (which could last up to 7 seconds), licking was rewarded with a 4–6 $\mu L$ drop of water, and caused an immediate transition to *ITI*. The *ITI* duration was uniformly distributed between 1 and 5 seconds, and afterwards the next trial began.

Beeps consisted of a harmonic chord composed of pure tones at 2, 4, 6, 8, and 16 kHz, presented via a speaker adjacent to the mouse at 70 dB SPL. The task was controlled using the Bpod behavior measurement and control system (Sanworks, USA). Licking behavior was monitored using a 200 fps camera (Blackfly BFS-U3-16S2M-CS, FLIR, USA) with a Tamron M118FM16 lens (Tamron, Japan). Licking events were detected online using Bonsai [75].

## Animal training

Two batches of mice were trained: the first batch (17 mice) was trained on all tested $\theta$-values (0.1-0.9), while the second batch (7 mice) was trained and tested on $\theta = 0.3$ and 0.7 only. Before varying $\theta$, mice were initially trained on the task with a single $\theta$, and Pavlovian rewards, in order to condition them to lick the water spout. *Safe* and *Unsafe* responses were still rewarded and punished, respectively. Initial training with $\theta = 0$ interfered with the later learning of $\theta > 0$, so $\theta$ was initially set to 0.2 or 0.3. These early sessions taught the task fundamentals by exposing mice to reward-predicting *safe go*'s and unrewarding *unsafe go*'s. We began varying $\theta$ once mouse behavior indicated that they had learned the association between *go* cue and reward.

The first batch of mice experienced 3–9 sessions in which $\theta$ values ranged from 0.1 to 0.7. Once mice demonstrated task acquisition by reliably responding to the state change, sessions with multiple $\theta$ values were introduced, presented in blocks of 30–100 trials. Mice were considered experts upon reaching stable performance, which occurred after ~10 training sessions. Upon achieving expert-level performance, we collected 10 sessions per mouse for analysis (Fig 2B and 2C).

The second batch of mice (Fig 2D and 2E) underwent sessions with $\theta = 0.3$ or 0.7 changing in blocks of 30-60 trials. The first 5 sessions with both contexts (sessions 6, 7, 8, 9, and 11; session 10 was excluded as it only included $\theta = 0.3$) were classified as 'novice' data. The last 5 training sessions (sessions 15–19) were classified as 'expert' data.

## Animal behavior analysis

Behavioral data were analyzed in MATLAB (MathWorks, USA). Trials were included in the analysis if the mouse licked at least once and if the trial occurred within the first 300 trials of the session. The first four trials of each session were excluded from adaptation analyses (Fig 2C–2E) because no prior context was available in the first block for mice to adapt their behavior relative to a preceding condition.

For correlation analyses, block averages were computed using all trials in a block except the first and last four trials. To examine the relationship between waiting time and block transitions (Fig 2C), we calculated the Pearson correlation between waiting times of individual trials and the mean waiting time of the preceding or current block.

Waiting time differences (Fig 2D) were computed as the average first-trial waiting time in $\theta = 0.7$ minus the average first-trial waiting time in $\theta = 0.3$. Two-sided Wilcoxon signed-rank tests were used to compare first-trial waiting times (N = 7 mice) (Fig 2E). Pearson's correlation coefficients and significance values were computed using MATLAB's corrcoef function. Error bars indicate standard error across mice.

## Neural network training

**Architecture.** We used the PyTorch library to implement an Actor-Critic framework [76]. Both the Actor and the Critic consisted of an LSTM layer (100 hidden units) followed by a linear readout. The readout of the critic was a scalar value, $V_t$, which was used to compute the Advantage $A_t = \gamma G_{t+1} + R_t - V_t$ where $\gamma$ is a discount factor, $R_t$ is reward, and $G_t = \sum_{t'=t}^{T} \gamma^{t'-t} R_t$ is the discounted future return. As per the standard Advantage Actor-Critic (A2C) algorithm, $V_t$ was used solely to train the actor. The actor readout produced 2 pre-activations, which we termed $Q$(action) and $Q$(inaction). The 2 pre-activations were passed through a softmax to yield action with probability:

$$P(action) = \frac{1}{1 + e^{-Q_t}}$$

where $Q_t = Q$(action) $- Q$(inaction) is the network decision variable. For standard policy execution, we sampled from the softmax distribution to facilitate trainability; when comparing to the optimal Bayesian agent, actions were chosen by

argmax:

$$P(action) = \begin{cases} 1 \ (\text{act}), & Q_t > 0, \\ 0 \ (\text{don't act}) & \text{otherwise,} \end{cases}$$

because the Bayesian policy is deterministic as well given a belief state.

**Input and initialization.** At each time step, both LSTMs received four inputs: (i) a scalar representing the current go/no-go state, (ii) the previous action, (iii) the previous reward, and (iv) optionally, the scalar context $\theta_t$ (for the untrained + context variant). Weight parameters were initialized using PyTorch's default linear layer initialization, and LSTM hidden states were zero-initialized before the first episode. For subsequent episodes, hidden states were carried over from the final time step of the preceding episode to avoid signaling the start of a new episode.

**Training procedure.** Networks were trained for 10,000 episodes, each comprising 20 trials under a single $\theta$ (sampled from 20 equally spaced values in [0, 1]). Each trial ended when the network responded. We applied a standard policy-gradient update with an entropy bonus of 0.05 to the actor, while the critic loss was scaled by 0.01 relative to the Actor [55]. Critic weights were updated by minimizing the advantage. The explicit loss of the critic was $loss_{critic} = 0.01 \sum_{t=0}^{T-1} (\gamma G_{t+1} + R_t - V_t)^2$, with a discount factor $\gamma = 0.5$. The final return $G_T$ was set equal to $V_T$. The policy-gradient (with entropy bonus) loss applied to the actor was $loss_{actor} = \sum_t - \ln(\pi(a_t)) A_t + 0.05 \times \pi(a_t) \ln(\pi(a_t))$ Optimization used RMSprop with an initial learning rate of 0.0005, decaying by 0.99 every 10 episodes, and momentum/weight decay both set to 1e-3. Gradients were computed via backpropagation through time at the end of each episode, then detached.

**Task and reward structure.** The *unsafe* state duration was drawn from an exponential distribution with a mean 10 steps (minimum of 1), always beginning with a forced no-go step. The intertrial interval (ITI) was uniformly distributed between 5 and 25 steps. **During training**, correct actions in the *safe* state delivered a reward of +1, whereas acting during the stimulus period incurred a negative reward penalty ($r = -3$) to stabilize learning. At inference time, this negative reward was removed.

**Evaluation.** We trained 10 networks per model variant (trained RNN, untrained RNN, and untrained RNN + context). Variability across the 10 networks could result from stochasticity in the weight-initialization, state durations, stimuli, and policy. After reaching comparable performance across all networks, the weights were frozen, and 5,000 episodes per network were evaluated and pooled for the final behavioral analysis.

### Analysis of network data

**Principle component analysis.** PCA was performed on the Short-Term Memory stream of the Actor LSTM. Data from all units across all trails (including intertrial intervals) were centralized and used to compute the principal components using Python's *scikit-learn* PCA package.

**Fitting network output to the linear integrator model.** To model the output of the recurrent network over time $Q_t$ as a linear integrator, we optimize the model parameters under constraints $0 < \alpha < 1$, $\beta > 0$, and $\epsilon_0 > 0$. Optimization was performed using the first 100,000 data time steps, while the entire dataset was used to evaluate goodness of fit (Fig 4G and 4H). The variability between the network realizations ($n = 10$) was negligible and did not affect the estimated fit parameters.

**Drift-diffusion models.** To compare our results with conventional evidence-accumulation approaches, we implemented a leaky integrator to estimate $\theta$:

$$\hat{\theta}_t = \alpha \hat{\theta}_{t-1} + (1 - \alpha) x_t.$$

and an integrator with reset inferred state according to

$$\hat{s}_t = \begin{cases} s_{t-1} + \frac{1}{\tau^\star(\hat{\theta}_t)}, & x_t = 1, \\ 0, & \text{otherwise.} \end{cases}$$

Performance was evaluated for $\alpha \in$ $\{0.85, 0.875, 0.9, 0.925, 0.95, 0.96, 0.97, 0.98, 0.99, 0.991, 0.992, 0.993, 0.994, 0.995\}$. 5000 episodes were collected for each agent.

## Vector field analysis

In order to measure the history and cue dependent update on each agent's decision variable and uncertainty, we randomly planted trials of a specific stimulus. The exact stimulus was $x_{0:20} = [0, 1, 1, 1, 1, 1, 1, 1, 1, 0, 0, 0, 0, 0, 0, 0, 0, 0, 0, 0]$. To avoid the confound of different policies on our analysis we prevent the agents from acting during the planted stimulus. This enabled us to collect an equal amount of data (approximately 500 trials per $\theta$) across agents and values of $\theta$ without the risk of selection bias. For each step of the planted stimulus we compute the change in each direction by taking the difference along each axis during and after the cue.

**Regression from network activity.** To decode ground truth and Bayesian estimates, we performed regression on the short-term memory stream of the actor network, as it represents neural activations relevant to decision-making. The critic network, while essential for learning, does not contribute significantly during inference and was therefore excluded [55].

All regressions were performed on trial data, excluding inter-trial intervals (ITI), as they are not behaviorally relevant. To obtain a linear readout, we applied logistic regression to decode the true state $s_t$ and the Bayes optimal state estimate $\hat{s}_t$, with the latter regressing to its log-likelihood ratio:

$$S_{\text{LLR}} = \ln\left(\frac{\hat{s}_t}{1 - \hat{s}_t}\right).$$

Context decoding was performed using linear regression to predict the true context $\theta_t$ and the Bayesian estimate $\hat{\theta}_t$. To decode the Bayesian decision variable, we regressed the network activity to the Bayesian state estimator $\hat{s}_t$ and the belief threshold $\hat{s}_t^*$ using the log-likelihood ratio, then calculated the decision variable as $\delta_t = \hat{s}_t - \hat{s}_t^*$. All regression models were trained on half of the trial data and tested on the remaining half.

## Bayesian agent

The Bayesian agent was numerically simulated by performing a sequential update of the joint posterior distribution $\mathcal{P}_t(s, \theta | \boldsymbol{x}_{\leq t})$ at each time step, following Eq (4). The agent was assumed to be Bayes-optimal, possessing perfect knowledge of the world model, including the observation likelihood $p(x_t | s_t, \theta_t)$ and the latent transition probabilities $P(s_t | s_{t-1})$ and $P(\theta_t | \theta_{t-1})$.

The agent initiated an action when the state estimate

$$\hat{s}_t = \sum_s s \, \mathcal{P}_t(s, \theta | \boldsymbol{x}_{\leq t}),$$

exceeded the optimal decision threshold,

$$\hat{s}_t^\star = \mathbb{E}_{\theta_t} \left[ P(s_t = 1 | \boldsymbol{x}_{t-\tau^\star:t} = 1, \theta_t) \right].$$

The optimal action time $\tau^\star$ was determined by maximizing the expected reward,

$$\tau^\star = \operatorname{argmax}_\tau \mathbb{E}_{\theta_t}\left[r(\tau;\theta_t)\right].$$

When the state transitions were determined by the RNN we approximated $\hat{s}_t^\star$ using $\hat{\theta}_t$ for simplicity.

## Supporting information

**S1 Text. Contains supplemental text, methods and figures.**
(PDF)

## Acknowledgments

We thank Alkesh Yadav, Vladimir Shaidurov, and Lior Fox for helpful discussions.

## Author contributions

**Conceptualization:** John Schwarcz, Jan Bauer, Eran Lottem, Jonathan Kadmon.

**Data curation:** John Schwarcz, Haneen Rajabi, Gabrielle Marmur, Eran Lottem.

**Formal analysis:** John Schwarcz, Jan Bauer, Jonathan Kadmon.

**Funding acquisition:** Eran Lottem, Jonathan Kadmon.

**Investigation:** John Schwarcz, Jan Bauer, Eran Lottem, Jonathan Kadmon.

**Methodology:** John Schwarcz, Jan Bauer, Eran Lottem, Jonathan Kadmon.

**Project administration:** Eran Lottem.

**Resources:** Eran Lottem, Jonathan Kadmon.

**Supervision:** Eran Lottem, Jonathan Kadmon.

**Validation:** John Schwarcz, Eran Lottem, Jonathan Kadmon.

**Visualization:** John Schwarcz, Jan Bauer.

**Writing – original draft:** John Schwarcz, Jan Bauer, Jonathan Kadmon.

**Writing – review & editing:** John Schwarcz, Jan Bauer, Gabrielle Marmur, Eran Lottem, Jonathan Kadmon.

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
