## [Decision Letter · Decision Letter 0]

7 Jun 2025

PCOMPBIOL-D-25-00425

Neural mechanisms of flexible perceptual inference

PLOS Computational Biology

Dear Dr. Kadmon,

Thank you for submitting your manuscript to PLOS Computational Biology. After careful consideration, we feel that it has merit but does not fully meet PLOS Computational Biology's publication criteria as it currently stands. Therefore, we invite you to submit a revised version of the manuscript that addresses the points raised during the review process.

Unlike Reviewer 3 suggestion, I think that the paper should not be split. I would like to stress that an effort should be made regarding the behavoural modelling, e.g. by comparing quantitatively alternative models as pointed out by the reviewers. 

Please submit your revised manuscript within 60 days Aug 07 2025 11:59PM. If you will need more time than this to complete your revisions, please reply to this message or contact the journal office at ploscompbiol@plos.org. Please include the following items when submitting your revised manuscript:

We look forward to receiving your revised manuscript.

Kind regards,

Bastien Blain

Academic Editor

PLOS Computational Biology

Joseph Ayers

Section Editor

PLOS Computational Biology

**Journal Requirements:**

2) <carina-action-element class="ng-star-inserted">Please provide an Author Summary. This should appear in your manuscript between the Abstract (if applicable) and the Introduction, and should be 150-200 words long. The aim should be to make your findings accessible to a wide audience that includes both scientists and non-scientists. Sample summaries can be found on our website under Submission Guidelines:</carina-action-element> 

<carina-action-element class="ng-star-inserted">https://journals.plos.org/</carina-action-element><carina-action-element class="ng-star-inserted">ploscompbiol</carina-action-element><carina-action-element class="ng-star-inserted">/s/submission-guidelines#loc-parts-of-a-submission</carina-action-element>  3) <carina-action-element class="ng-star-inserted">Please upload all main figures as separate Figure files in .tif or .eps format. For more information about how to convert and format your figure files please see our guidelines: </carina-action-element> <carina-action-element class="ng-star-inserted">https://journals.plos.org/</carina-action-element><carina-action-element class="ng-star-inserted">ploscompbiol</carina-action-element><carina-action-element class="ng-star-inserted">/s/figures</carina-action-element>  4) We have noticed that you have uploaded Supporting Information files, but you have not included a list of legends. Please add a full list of legends for your Supporting Information files after the references list.   

**Reviewers' comments:**

Reviewer's Responses to Questions

**Comments to the Authors:**

Reviewer #1: In the manuscript entitled “Neural mechanisms of flexible perceptual inference”, John Schwarcz and colleagues investigate the computational mechanisms required to solve an original perceptual decision task, involving the inference of both the current state and latent contextual statistics – coined flexible perceptual inference in the manuscript. The task was an auditory go-nogo with varying levels of contextual uncertainty: in a low uncertainty context go-signals were more likely to predict a correct go response, while in a high uncertainty context go-signals were more likely to be unreliable. Correct go responses lead to an immediate reward, while incorrect (premature) responses lead to no reward. Crucially, the agents didn’t know which context they were in, and had to infer the reliability of the sensory signal in each trial in order to make an accurate response.

The authors combined behavioral analyses in 2 groups of (head fixed and water-restricted) mice (n=17 and n=7), a derivation of the normative model of their task and analyses of emerging patterns in recurrent neural networks. Their main results are threefold: 1) mice displayed a non trivial behavioral pattern, showing fast adaptation of response times after a context change, even before receiving feedback, 2) the normative model solved the task by adjusting evidence weight and decision threshold at the same time and reproduced the main behavioral patterns found in mice, and 3) recurrent neural networks trained to maximize the rewards rate in the task behaved similarly to mice, with their recurrent activity correlating with latent variables of the optimal bayesian model.

The authors conclude that animals perform flexible perceptual inference by inferring contextual statistics as in the normative bayesian model, and that RNNs offer a plausible neural mechanism to implement such computations in biological agents.

Overall, the manuscript is well written and the presentation of results and arguments is very fluid. The main behavioral result, that mice adjust to latent statistics beyond feedback processing, is reminiscent of the seminal work of Gallistel and Gibbon (e.g. Gallistel and Gibbon 2000, Gallistel et al 2001), and well supported by their data. However, I'm not completely satisfied with several of the authors' claims:

Classical evidence accumulation models cannot account for context inference

Drift diffusion models are mentioned at multiple times in the manuscript, and the authors are well aware of their link with optimal perceptual decision making. In the subsection “Comparison with drift-diffusion dynamics” (line 358), the authors verbally compare the normative bayesian solution to so-called “standard DDM assumptions”, i.e. fixed decision boundary and drift rate. However, previous work cited in the manuscript show that normative solutions sometimes correspond to DDMs with collapsing boundaries (e.g. Tajima et al 2016) and that animal behavior can be accounted for by DDMs with variable and adaptive drift-rates (e.g. Brunton et al 2013). For these reasons, I disagree with the conclusion that “simultaneously inferring both state and context yields distinctive, context-dependent changes in the decision-variable dynamics that break the conventional logic of drift-diffusion models.” (lines 392-393).

Instead of discussing an unfairly restricted version of these models, a quantitative comparison of DDMs simulations with adaptive boundaries and drift-rate would directly test their ability to offer a mechanistic interpretation of behavioral data.

RNNs as plausible neural mechanistic models

The authors’ use of RNNs is twofold: 1) as an algorithmic implementation of the normative bayesian model, following the idea that drift-diffusion models are not suitable in this case (but see my first point) and 2) as a plausible biological implementation.

They clearly show that a trained RNN converges to approximate the normative bayesian solution in their task at the superficial, behavioral, level, but also at the underlying level of latent decision variables. This result is not surprising given the existing literature on the ability of these models to converge to the optimal solution in inference tasks. However, I think the authors overstate how these analyses inform us about biological computations for flexible perceptual inference in general.

First, while RNNs can resemble biological neural circuits, the fact that a trained RNN performs near-optimal Bayesian inference in a task does not necessarily make it a biologically plausible mechanism for such inference. This is because, in general, bayesian inference is intractable (even approximate bayesian inference, see Kwisthout et al 2011). Biological systems rely on limited resources, and despite being able to perform computations that mimic normative bayesian inference in some instances, this cannot be guaranteed a priori. For example the statement “while Bayesian inference requires maintaining and updating joint probability distributions over latent variables, the network has discovered an efficient solution that captures the essential dynamics of the decision variable updates.” (lines 499-500) does not hold in general, since there is no “efficient solution” for an intractable problem. Similarly the statement “By training RNNs on such tasks, we uncover neural mechanisms underlying flexible perceptual inference” (line 659) goes too far. Since the RNN reached a near-optimal behavior in this task by using representations similar to those of the normative model, it gives no additional information than the normative model about how biological systems could solve the task, not to mention solving flexible perceptual inference in general.

Second, in order to fully complete the bridge between the normative model and biological system, the authors don’t provide biological neural activity matching that of artificial networks (as in e.g. Sohn et al 2019). This is not a major issue in the scope of their study, but the title is misleading. Moreover, the statement: “This approach bridges theoretical models and experimental neuroscience, offering a powerful framework for studying the mechanisms underlying complex cognitive functions” (lines 660-661) is more a case of wishful thinking than actual demonstration. Furthermore, while they convincingly show that training the recurrent weights is necessary for matching animal and normative performance, even when explicitly knowing the context, they again underexplain why this is the case. “In contrast, trained recurrent networks (Fig. 5f) develop internal dynamics that independently control both integration rate and baseline in the precise and complementary manner predicted by Bayesian theory.” (lines 598-600) Either the RNN approximates the normative Bayesian model, and therefore offers no additional insight, or it implements a sufficiently close heuristic, which needs to be made explicit (e.g. a dynamically adaptive DDM ?)

To summarize, the authors should deflate their claims about biological plausibility/neural underpinnings (also in the title), and/or give a more in-depth investigation into how neural networks increase our understanding of inference mechanisms with respect to the normative model.

Overstatement about metalearning

Regarding the claim that flexible perceptual inference is learned, the authors write: “From a computational perspective, our work extends the framework of meta-learning—the concept of learning to learn—to encompass learning to infer.” (lines 631-633). This is a strong overstatement since no analysis in the manuscript actually involves learning the adaptive inference mechanisms. The only contribution on this matter is in the behavioral data of the second group (n=7) showing a training effect in mice (naive mice have a lower early contextual adaptation effect than expert mice). While this result is very interesting and indeed suggests that animals learned to infer correctly both the current state and context, all the subsequent analyses involve models with fixed parameters throughout the task. Here, the internals of RNNs might be revealing: since RNNs are not constrained by specific algorithmic assumptions, fitting an RNN to the responses of the animals (and not rewards, see e.g. Eckstein et al 2024) at early and late stages of training could give an insight of how relevant computations are learned. Also, all mice were trained in low uncertainty contexts first (theta = 0.2/0.3). I understand why it’s better (and maybe necessary) for the animal to actually learn the task, but it can bias their overall performance (in contrast, the RNNs were trained with all possible contexts). If training with higher initial uncertainty was tested and did not allow the animals to properly learn the task, it should be mentioned.

Minor points:

- The labels in fig. 2c (“current theta”/“previous theta”) must be reversed

- The figures order doesn’t always follow how they appear in the main text, which disturbs readability (e.g. figure 5d is mentioned before figures 5a,b and c)

- line 783 : “Intgrator” instead of “Integrator”

- The red dashed lines in figures 3f, 4e, 4h, and 5h should be labeled. Especially in figure 5 it is not clear to me what they represent.

- The term “model-based” appears twice in the manuscript (lines 626 and 648), without an explicit definition of what it means in the context of this study. Since it’s an established terminology in the reinforcement learning literature, referring to the representation of transition functions that are not the scope of this study, I recommend to avoid using this term.

Reviewer #2: Comments to Authors

In this study, Schwarcz, Kadmon and colleagues study, using both theory and experiments, normative approaches to the problem of context-dependent perceptual inference. They first develop a novel variant of the ‘integrate and reset’ family of tasks (e.g., ref 61) and show that mice adapt integration/waiting times as a function of the unreliability (‘false alarm’ probability) of the environment and also that they change policies in the first trial after each block change without reinforcement, which suggests that the mice understand the generative model of the task. Next, they derive the optimal policy for this task using POMDPs, and finally they analyze the extent to which RNNs trained in the task (and various variants), use strategies consistent with the normative solution.

The study, which is rigorous, thorough and elegant, describes generic principles relevant for our understanding of this timely and important problem. I congratulate the authors for their accomplishment.

Comments:

Experiments

1) The task shares with previous instantiations the informational asymmetry of the two possible observations with respect to the latent state. An essential (and useful) aspect of this asymmetry is that it provides a simple qualitative readout of whether the subject has learned the generative model: integration should be reset with each observation of, in this case, the nogo stimulus (because the prob of observing a nogo in the safe state is zero). Do mice show evidence of such resetting behavior? The authors show that waiting times increase with theta, but they do not show whether waiting time is, on average, independent of how many go stimuli were presented before the last nogo.

2) Is it possible that the mice use a heuristic whereby waiting time is an increasing function of the running average of go stimuli within a trial? In this case, the difference between novice and expert mice might be how deterministic this function is. The optimal strategy can be described in this way, but it would seem like various suboptimal/heuristic strategies might also have this property. The authors nicely call attention to the difficulty of identifying whether their RNN agents are indeed using the optimal strategy, and it seems like even greater difficulties would apply for real agents where we don’t have access to the majority of the relevant internal signals. Along this lines, perhaps the plots suggested in point (1) would provide further evidence that the mice indeed approach the optimal strategy.

Theory

1) The authors employ a different approach from previous works who have used POMDPs for deriving the optimal policy in (different) perceptual decision tasks (e.g., Moreno, Drugowitsch et al., 2012 and ref 12). If I understand correctly, they first compute the optimal waiting time by maximizing the average (over the reliability parameter theta) of an expression for the reward rate conditional on theta. From here, they derive a decision bound by evaluating the average (over the reliability parameter theta) of the GO probability at the optimal waiting time. But the derivation of the optimal waiting time does not depend explicitly on any ‘policy’. Only implicitly, I guess, through the fact that the (average) reward rate after the optimal waiting time has been exceeded. I might be wrong, but it doesn’t seem trivial to me that the behavior produced by using this policy would be the same as the behavior resulting from a policy that consists in applying, moment by moment a hard bound on the task relevant belief (which is the approach that was followed in previous studies). This is because the belief trajectories in the presence of a bound will not necessarily coincide with those used for computing the various averages in the method used by the authors. Perhaps the authors could clarify this issue.

I think discussing a bit more in detail the similarities/differences between the POMDP solution of the standard 2AFC task in the refs above and the one described here would also be nice.

2) Are the bounds on belief time dependent? I could not clearly stated whether this is the case except in the discussion in line 614 “bounds dynamically adjust in real time”. It seems to me like they should depend on time because they depend on theta and the agent’s estimate of theta is evolving (at least after a block change), but I might be wrong. This seems quite important so whether they are constant of time-varying should be stated and explained more explicitly.

Another reason to emphasize this is that the decision variable is defined as the distance from the momentary belief to the bound. If both are time-varying, then it would be nice to plot them separately under different conditions.

I think it will help the reader to see simulations of single-trial belief trajectories in addition to the averages in e.g., Fig3f. These could be aligned to the action time, like in the figure, or on trial onset. Time-varying theta estimates after a block change would also be nice to see to gain intuition. Generally speaking, more plots describing the dynamics of all relevant quantities would, I think, help the reader understand the solution found by the authors.

3) Rigorously speaking, the reward-rate maximizing policy should include, in the denominator the consequences of mistakes where the animal licks in the unsafe state (7 sec penalty). Probably the qualitative features of the solution do not change adding this feature, and thus I don’t think it’s necessary that the authors compute this new policy, but maybe they can comment on this in the discussion.

Networks

1) I could not fully understand what the cost function was for the networks. In lines 461-62 it says “networks were rewarded for implementing appropriate waiting times”. In the methods (lines 762, 764) the cost is given for the critic, but not explicitly for the actor. Would be nice to clarify.

2) The authors argue that “The network could have adopted simpler strategies, such as basic evidence accumulation with context-dependent thresholds. Instead, it discovered how to approximately mirror the complex, context-dependent dynamics of optimal sequential inference, including precisely calibrated integration rates and initial conditions” (lines 501-504). I’m not sure I understand this reasoning. This seems to suggest that those alternative solutions would be equivalent from the point of view of the cost function? Easier to learn (inductive bias)?

Minor points/typos

- line 167: see supplementary. Couldn’t find anything related to this in the supplementary.

- line 246: on -> when conditioned on

- line 275: I couldn’t find a detailed derivation of Eq. 4 on the SupInfo.

- Fig4g. Please be more explicit about the legend. Wasn’t sure what p(x) referred to.

- Section Architecture (line 748): Please be more explicit about the relationship between the value V and Q.

- Calculation of reward rate in SupInfo: It was hard for me to understand this (critical) derivation until I read ref 76. For completeness and for the manuscript to be more self contained, I would encourage the authors to unpack this a bit (as is done in the reference) to make it easier for the reader to see how this was done.

- SupInfo line 1072: a -> lambda?

Reviewer #3: I would like to thank the authors for this nice study tackling a fundamental problem in decision making which how agents flexibly adjust their decisions depending on the context. The authors trained mice to do the a task in which they infer state and context changes which is open up a lot of avenues of behavioral and neural mechanisms.

First, i would like to raise some general conceptual comments that I would like the authors to think about :

1. I think the behavior part by itself can be a standalone paper which entails removing the part in which you simulate neural networks generating the neural activity that can underlie the behavioral dynamics you observe. You have really nice behavioral data and you are able to understand it deeply using quantitive modellng, which constitute a study by itself. I would rather to see the neural network part in another paper along with neural recordings during the task. In any case this is a recommendation, i am totally fine if you decide to keep it.

2. Relating to the previous point, the authors can extend the behavioral modeling by simulating the DDM models they are proposing or at least propose a functional form that you think might be underlying the modified DDM model. I would also recommend that you try using a two or multiple accumulator models that can reproduce the behavioral data. We lack in the literature studies that compare POMDPs and DDMs on the same data set, so this might be your opportunity to give us insights into this. Moreover, it will be great to speculate on how can we differentiate between them, may be it is here when the neural mechanisms become crucial.

Second, I have comments concerning the details of the study:

1. Figure one needs to improved signifcantly. Fig 1b needs to include the behavioral task design in a more details like in many other mice or rats behavioral papers eg. including the range of the trial length, the values of theta, etc. (I read them later in fig 2B and the Methods but they should be in fig 1 for clarity of the design.) just so that the reader can quickly get the quantitative aspects of the behavioral design. Moreover, I have not seen the psychometric throughout the whole paper including the supplement. I think including a psychometric curve in fig 1 important to show the overall performance of the mice. It was hard for me to gauge how good they are at doing the task ; is it 100 or 60 percent ? I am also curious about the terminology 'safe' or 'unsafe' you can use State 1 and State 2 without using a terminology that might imply that the animal in itself is feeling safe or not (like in a predator - prey context or with looming stimuli).

2. I understand that mice adapt from the very first trial after switch but i see in fig 2 c that the correlation seems to be increasing so can you clarify the values of what one can call timescale of adaptation. My impression, correct me if I am wrong, is that when the switch happens they are performing the task with okay accuracy then better then better, does the performance plateau afterwards ? It is great that you did the experiment with novice versus expert and what I find interesting is that although the switch of the novie is gradual and the expert is quick, to my eyes they reach similar performance after three trials nevertheless they exhibit asymmetry in their behavior during high uncertainty context, what's your interpretation for this ? can this be accounted for using your RL driven RNN training.

3. Fig 3, the panels are too small, please increase their size.

Third, a minor comment, in line 150 " to inform their... " there is a missing word after their.

Overall, I like the paper very much but I would recommend you address the aforementioned comments before it is considered for publication.

**Have the authors made all data and (if applicable) computational code underlying the findings in their manuscript fully available?**

Reviewer #1: **No: **All scripts are publicly available, but the data is not, and only provided upon request to the authors.

Reviewer #2: None

Reviewer #3: **No: **There is only link to the modelling of neural networks but not of the behavioral analysis or the behavior data itself.

PLOS authors have the option to publish the peer review history of their article (what does this mean?). If published, this will include your full peer review and any attached files.

Reviewer #1: No

Reviewer #2: No

Reviewer #3: **Yes: **Ahmed El Hady

**Figure resubmission:**
---

## [Decision Letter · Decision Letter 1]

30 Sep 2025

PCOMPBIOL-D-25-00425R1

Neural mechanisms of flexible perceptual inference

PLOS Computational Biology

Dear Dr. Kadmon,

Thank you for submitting your manuscript to PLOS Computational Biology. The paper is now nearly ready to be accepted for publication but I believe the minor issues pointed out by reviewer 2 could be adressed easily. 

Please submit your revised manuscript within 30 days Nov 30 2025 11:59PM. If you will need more time than this to complete your revisions, please reply to this message or contact the journal office at ploscompbiol@plos.org. Please include the following items when submitting your revised manuscript:

We look forward to receiving your revised manuscript.

Kind regards,

Bastien Blain

Academic Editor

PLOS Computational Biology

Daniele Marinazzo

Section Editor

PLOS Computational Biology

**Journal Requirements:**

1) We note that your Supplementary Figures files are duplicated on your submission as they are included in the supporting information file and uploaded as separate files.Please ensure that there are uploaded only once with the file type 'Supporting Information.'

**Reviewers' comments:**

Reviewer's Responses to Questions

Reviewer #1: The authors have substantially improved the manuscript and my comments have been sufficiently addressed.

I spotted 2 typos :

line 495 a word seem missing : “after a transient...”

line 633 : “Baysian”

Reviewer #2: The authors addressed most points I had raised

I have one follow up question. The analysis in Figure 5 is revealing and nice. But it shows that the network does not reset upon experiencing a nogo stimulus. The authors say this is because the dynamics of the network is discrete (lines 506-7). I’m not sure I understand why this is the case (elaborate?) Either way, resetting is the essential component of the inference strategy. The network seems to gradually convince itself that the state switches to an unsafe state after several nogo cues, which is puzzling given the switching dynamics in Fig. 4c.

If this is a numerical issue, then It seems to me like the authors should decrease the time step of the simulations to show that the network can indeed reset. If it’s not numerical, then I guess it is worth understanding why reset is not there.

Also, I think it would be useful if the authors paired each panel with a time-course of the decision variable for those particular parameters. This will make it easier to the reader to link the 3 critical quantities: posteriors on state and uncertainty and DV.

Reviewer #3: I would like to thank the authors for the thorough revision and response to my previous comments. I have no further comments and I recommend the paper for publication.

**Have the authors made all data and (if applicable) computational code underlying the findings in their manuscript fully available?**

Reviewer #1: None

Reviewer #2: None

Reviewer #3: None

PLOS authors have the option to publish the peer review history of their article (what does this mean?). If published, this will include your full peer review and any attached files.

Reviewer #1: No

Reviewer #2: No

Reviewer #3: **Yes: **Ahmed El Hady

**Figure resubmission:**
---

## [Editor Report · Decision Letter 2]

28 Oct 2025

Dear Dr. Kadmon,

We are pleased to inform you that your manuscript 'Neural mechanisms of flexible perceptual inference' has been provisionally accepted for publication in PLOS Computational Biology.

Best regards,

Bastien Blain

Academic Editor

PLOS Computational Biology

Daniele Marinazzo

Section Editor

PLOS Computational Biology

---

## [Editor Report · Acceptance letter]

PCOMPBIOL-D-25-00425R2

Neural mechanisms of flexible perceptual inference

Dear Dr Kadmon,

I am pleased to inform you that your manuscript has been formally accepted for publication in PLOS Computational Biology. Your manuscript is now with our production department and you will be notified of the publication date in due course.

With kind regards,

Judit Kozma
